# Patterns of plant rehydration and growth following pulses of soil moisture availability

Andrew F. Feldman[1], Daniel J. Short Gianotti[1], Alexandra G. Konings[2], Pierre Gentine[3], Dara Entekhabi[1]

[1]Department of Civil and Environmental Engineering, Massachusetts Institute of Technology, 15 Vassar St., Cambridge, Massachusetts, 02139, USA
[2]Department of Earth System Science, Stanford University, Stanford, California, USA
[3]Department of Earth and Environmental Engineering, Columbia University, New York, New York, USA

*Correspondence to*: Andrew F. Feldman (afeld24@mit.edu)

**Abstract.** Plant hydraulic and photosynthetic responses to individual rain pulses are not well understood because field experiments of pulse behaviour are sparse. Understanding individual pulse responses would inform how rainfall intermittency impacts terrestrial biogeochemical cycles, especially in drylands which play a large role in global atmospheric carbon uptake interannual variability. Using satellite-based estimates of predawn plant and soil water content from the Soil Moisture Active Passive (SMAP) satellite, we quantify the timescales of plant water content increases following rainfall pulses, which we expect bear the signature of whole-plant mechanisms. In wetter regions, we find that plant water content increases rapidly and dries along with soil moisture, which we attribute to predawn soil-plant water potential equilibrium. Global drylands, by contrast, show multi-day plant water content increases after rain pulses. Shorter increases are more common following dry initial soil conditions. These are attributed to slow plant rehydration due to high plant resistances using a plant hydraulic model. Longer multi-day dryland plant water content increases are attributed to pulse-driven growth, following larger rain pulses and wetter initial soil conditions. These dryland responses reflect widespread drought recovery rehydration responses and individual pulse-driven growth responses, as supported by previous isolated field experiments. The response dependence on moisture pulse characteristics, especially in drylands, also shows ecosystem sensitivity to intra-annual rainfall intensity and frequency, which are shifting with climate change.

## 1. Introduction

A changing climate is likely to shift not only mean annual rainfall, but also the frequency and intensity of rainfall events (Donat et al., 2016; Giorgi et al., 2019; Trenberth, 2011). Understanding the impacts of rainfall distribution shifts on the terrestrial biosphere is important because vegetation globally exerts a large control on terrestrial water and carbon balances (Ahlström et al., 2015; Jasechko et al., 2013; Poulter et al., 2014) and drives feedbacks with the lower atmosphere (Gentine et al., 2019; Green et al., 2017). Changing rainfall frequency with the same annual rainfall can impact terrestrial carbon uptake (Fay et al., 2003; Knapp et al., 2002), suggesting ecosystem sensitivity to characteristics of individual rain events. This motivates characterizing plant responses to individual moisture availability pulses across climates and biomes. This is especially the case for semi-arid herbaceous plants which respond primarily to individual pulses, likely occurring under a pulse-reserve paradigm of individual rainfall events triggering photosynthetic responses and storages (Collins et al., 2014; Feldman et al., 2018; Noy-Meir, 1973). However, vegetation responses on these shorter timescales are less well understood than the more commonly studied responses to monthly or annual water anomalies.

A major component of understanding moisture pulse responses is quantifying the duration over which plants take up and use the rain pulse water. These response durations bear the signature of, among others, plant rehydration in the soil-plant atmosphere continuum (SPAC), growth, and resource limitations such as drought (Manzoni et al., 2013; Ogle et al., 2015; Ogle and Reynolds, 2004; Sperry et al., 2016). Characterizing these timescales and their dependencies across biomes will

increase our understanding of whole-plant behaviour and assist plant hydraulic parameterizations in land surface models to better assess plant water stress (Bonan et al., 2014; Fisher et al., 2018; Kennedy et al., 2019; Lin et al., 2019; Tai et al., 2017; Xu et al., 2016). However, the fundamentals of these pulse water use durations, especially soil to plant water storage timescales, remain unknown globally.

Soil and plant water content measurements can be used to characterize the soil-plant hydraulic system and understand plant water storage timescales, but are laborious and are thus often constrained to a single location. Alternatively, microwave remote sensing satellites provide plant water content observations across the globe at near-daily sampling frequencies (Entekhabi et al., 2010; Kerr et al., 2010; Konings et al., 2016). Although these observations are at a coarse resolution (10s of kilometers), their ecosystem scale resolution and relationship to leaf water potential (Momen et al., 2017; Zhang et al., 2019)

makes them a useful tool for studies of ecosystem plant-water relations (Feldman et al., 2018; Konings et al., 2019; Konings and Gentine, 2017). However, the plant water content measurements are a function of both relative water content and dry biomass, thus making them additionally sensitive to biomass and growth (Momen et al., 2017; Zhang et al., 2019). This is nevertheless an advantage because the plant water content sensitivity to both water potential and dry biomass allows evaluation of timescales of multiple whole plant mechanisms at the landscape-scale.

With regard to plant rehydration timescales, plant water uptake and storage timescales have been typically assessed using an electric circuit analogy, with the timescale of interest the plant resistance times capacitance, or RC time constant (Phillips et al., 1997, 2004; Ward et al., 2013). The RC time constant quantifies the time required for leaf or xylem water potential to reach 63% of its equilibrium value following a soil moisture or transpiration perturbation. Measured RC time constants vary from minutes for grasses to hours for trees (Hunt and Nobel, 1987; Nobel and Jordan, 1983; Phillips et al.,

1997, 2004; Ward et al., 2013). According to this theory, plant rehydration after rain pulses should occur within a day. By contrast, field pulse experiments of grass and shrub species, primarily in dryland environments (broadly annual rainfall less than 500 mm), commonly show multi-day predawn water potential increases after rewetting pulses (Fravolini et al., 2005; Huxman et al., 2004; Ignace et al., 2007; West et al., 2007). Multi-day predawn water potential increases would lengthen plant water content timescales. These multi-day water potential increases appear to be related to recovery from water-limitation

between storms, which highlights the potential impact of antecedent moisture conditions on plant responses (Guo and Ogle, 2019; Ogle et al., 2015; Plaut et al., 2013). Hydraulic limitations from previously dry conditions have also been observed, driving multi-day recovery of leaf gas exchange after soil rewetting (Blackman et al., 2009; Brodribb and Cochard, 2009; Chen et al., 2009; Huxman et al., 2004; Martorell et al., 2014). These multi-day hydraulic response observations call into question whether hydraulic response timescales are consistently sub-daily across global biomes. They also highlight the unknown role

of moisture pulse characteristics (antecedent soil conditions and pulse magnitude) on these timescales.

With regard to timescales of growth, while growth is known to occur on seasonal timescales, there is evidence that growth can occur following rainfall events as under the pulse-reserve hypothesis (Noy-Meir, 1973). Specifically, dryland measurements suggest that growth can occur over days to weeks following a pulse (Angert et al., 2007; Dougherty et al., 1996; Hermance et al., 2015; Novoplansky and Goldberg, 2001; Post and Knapp, 2019; Sher et al., 2004). Also, ecosystem growth

responses in drylands have been modeled previously with a 1-5-day lag (depending on plant type) and a decaying persistence over 1-2 week scales (Ogle and Reynolds, 2004; Reynolds et al., 2004). Ultimately, pulse driven growth following rainfall would lengthen plant water content timescales by increasing the total plant water storage capacity.

Here, we evaluate the duration of total plant water content increases following rainfall pulse events. Under nominal moisture conditions with no growth, one would expect sub-daily plant water content increases based on RC time constants.

Slow rehydration and/or growth would likely extend these timescales to multiple days. We ask: across global biomes, do plant water uptake responses to soil moisture pulses ever occur beyond a day and what are these timescales? How do pulse characteristics (pulse magnitude, moisture pre-conditions) and growth influence these timescales? Do attributes of the moisture pulse (pulse magnitude, moisture pre-conditions) favor plant growth versus rehydration? To address these questions, we use microwave remote sensing of total plant water content, a combination of dry biomass and relative water content, following an

approach for rain pulse studies originally developed in Feldman et al. (2018). In order to better understand potential mechanisms underlying remote sensing observed timescale variations, we also discuss the observed timescales and their drivers in the context of a SPAC model.

## 2. Methods

**2.1 Datasets**

We use four years (April 1st, 2015 to March 31st, 2019) of soil moisture and plant water content observations from the Soil Moisture Active Passive (SMAP) satellite (Entekhabi et al., 2010). SMAP measures the low frequency microwave (1.4 GHz) radiation emitted from Earth's surface. The radiation signal is in units of temperature, or brightness temperature (TB). The radiation is polarized, where the emitted waves' oscillations have distinct horizontal ($TB_H$) and vertical ($TB_V$)

orientation. SMAP measures both $TB_V$ and $TB_H$. Both $TB_V$ and $TB_H$ magnitudes alone are sensitive to surface soil moisture (top ~5 cm). Furthermore, the difference between $TB_V$ and $TB_H$ is sensitive to how much the emitted waves are attenuated when traversing a vegetation canopy. The vegetation attenuation of the microwave radiation is called vegetation optical depth (VOD). More vegetation water content results in higher VOD (Jackson and Schmugge, 1991; Konings et al., 2019). An established radiative transfer equation can partition the $TB_V$ and $TB_H$ signals into soil moisture and VOD (Mo et al., 1982;

Wigneron et al., 2017). We use a recently developed algorithm, called the multi-temporal dual channel algorithm (MT-DCA), to robustly estimate soil moisture and VOD using this radiative transfer equation (Konings et al., 2017, 2016).

The SMAP satellite measurements occur at 6:00 AM (local time) everywhere on 9 km grids across the globe. 6:00 AM is approximately predawn, when plant water status is assumed to be maximal (due to nighttime plant rehydration). The satellite orbit is such that there is a one-, two-, or three-day revisit, depending on the day and latitude. Due to the orbit pattern,

higher latitudes are measured more frequently. This results in sampling frequencies of 1-2 days at midlatitudes and 2-3 days at the equator.

Since VOD has been shown to be nearly linearly proportional to total vegetation water content (Jackson and Schmugge, 1991), VOD is proportional to the product of relative water content and aboveground dry biomass (Konings et al., 2019; Momen et al., 2017; Zhang et al., 2019). Therefore, VOD can increase due to either rehydration of cell water storages

or biomass growth, as growth provides additional water storage capacity. VOD is expected to be sensitive to rehydration because of near-linear relationships between relative water content and plant water potential, especially for herbaceous species which are primarily investigated in this study (Jones, 2014; Jones and Higgs, 1979; Konings et al., 2019; Nobel and Jordan, 1983). While the low-resolution of VOD estimates hinders species-specific or stand-scale assessments, it provides the opportunity to assess integrated, landscape-scale vegetation behaviour across global biomes (Feldman et al., 2018; Tian et al.,

2018). VOD shows promise for use in monitoring plant water stress with recent findings showing VOD can monitor time evolution of plant water stress and drought-induced mortality with loss of plant water storage (Feldman et al., 2020b; Martínez-Vilalta et al., 2019; Rao et al., 2019).

Soil moisture observations from the MT-DCA algorithm compare closely to other SMAP soil moisture products (which use different algorithms) as well as to in-situ observations (Chan et al., 2016; Dadap et al., 2019; Feldman et al., 2018).

Direct in-situ VOD information is unavailable, although SMAP VOD's mean and dynamics are comparable to another satellite VOD product (Kerr et al., 2010). For further discussion of SMAP VOD estimate performance and comparison with other products, we refer the reader to Konings et al. (2017) and Feldman et al. (2018).

To assist in discriminating VOD changes related to hydraulic or growth activity, we use the daily leaf area index (LAI) product from the Spinning Enhanced Visible and Infrared Imager (SEVIRI) on-board EUMETSAT's Meteosat Second

Generation (MSG-2) satellite series (Trigo et al., 2011). These LAI observations serve as an indicator for above-ground biomass independent of VOD. While constrained primarily to Africa, these LAI observations are estimated from 15-minute geostationary observations which provide daily LAI fluctuations after cloud contamination mitigation. Both VOD and LAI datasets together are required to determine the occurrence of pulse driven growth. VOD increases can be linked directly to a

specific rain event because of SMAP's more rapid effective sampling (due to no cover contamination), but are confounded by rehydration. LAI changes over the weekly scales of pulses can detect canopy growth, but because of a non-linear averaging technique (García-Haro and Camacho, 2014), the LAI dataset is partially smoothed over sub-weekly scales and may be less apt to determine whether detected growth over a week is specifically associated with a given rain event. Nevertheless, increasing LAI over a rain event can identify whether VOD increases associated with that storm are due to growth or only rehydration. As such, we are interested in using LAI changes qualitatively to determine whether LAI is increasing or decreasing over more than week-long periods. Therefore, biases in LAI magnitude do not influence the analysis.

Use of SEVIRI LAI for this application is preferred due to SEVIRI's frequent sampling and filtering techniques that provide better resolution of the seasonal growth and senescence stages, especially during the wet season, than other available satellites (García-Haro et al., 2013; Gessner et al., 2013). SEVIRI ultimately provides 3-5 day effective sampling during cloud contaminated periods, which is typically four times less than effective sampling with low Earth orbit satellites (i.e., MODIS) (Fensholt et al., 2006). Therefore, despite being global, low Earth orbit satellites are not used because they sample too coarsely in time for the applications here. Furthermore, SEVIRI LAI retrievals in the herbaceous biomes evaluated in Africa have the lowest retrieval errors (García-Haro et al., 2013). Therefore, SEVIRI LAI is likely to detect increasing biomass over one to two week periods. While no other adequate satellites exist for direct comparison with results here, the analysis was repeated with SEVIRI fraction of absorbed photosynthetically active radiation (FAPAR) observations, derived from different measurement frequencies than LAI, and similar results were obtained (Fig. S1).

Ancillary data are used to evaluate climate and biome dependencies of the findings. Specifically, we compute mean annual precipitation using the Global Precipitation Measurement IMERG product (Huffman, 2015) and tree cover from the Moderate Resolution Imaging Spectroradiometer (MODIS) (Dimiceli et al., 2015) to evaluate VOD behaviour across climate gradients. International Geosphere-Biosphere Programme (IGBP) land cover maps are used to remove frozen and bare ground (Kim, 2013). Tree cover is also used to remove densely vegetated forests where soil moisture and VOD satellite estimation are uncertain.

**2.2 Soil Moisture Pulse Identification**

VOD behaviour and response timescales are evaluated during soil moisture drydown periods that occur after rainfall (Fig. 1). Drydown periods, or soil moisture pulses, are defined as an increase in soil moisture of at least 0.01 $m^3/m^3$ followed by a drying period of at least four consecutive measurements (approximately 8-12 days). This approach is nearly identical to previous approaches (McColl et al., 2017; Shellito et al., 2018). To remove seasonal drydowns less associated with an individual rainfall event, drying periods of longer than twenty days are not included. Seasonal trends (periodic climatology) are removed from soil moisture and VOD time series during drydowns, while preserving the magnitude of initial conditions (Feldman et al., 2019). This procedure removes seasonal VOD growth trends to isolate short term increases associated with a given storm that are due to pulse-driven growth and/or slow rehydration. Note that differences persist in the literature on whether the rain event or plant response is defined as the pulse (Reynolds et al., 2004). Both soil and plant responses are discussed as pulses here.

**2.3 Vegetation Pulse Response Timescale Estimation and Analysis**

For infiltrating water to ultimately reach the leaf, it must typically percolate from the soil surface to the roots, pass through the root endodermis, and move up the xylem through the shoot to leaves (Mackay et al., 2015; Sperry et al., 2016, 1998). Given an identified soil moisture drydown period, the VOD response timescale, defined here as time-to-peak ($t_p$), is estimated as the time from the beginning of the soil moisture drydown to the first local maximum value of VOD (Fig. 1). After a period of water storage, plant water content loss occurs due to surface and atmospheric drying and warming, creating a peak in plant water content conditions that $t_p$ attempts to quantify (Feldman et al., 2020b). $t_p$ captures the aggregate rehydration and growth timescale during this soil-plant water transport process. The $t_p$ estimation relies on consecutive VOD increases, which

provide more robust estimates than the global maximum during the drydown. To increase sample size, we conduct the analysis on all pixels contained within a 0.5°x0.5° domain (includes ~30 SMAP pixels). Densely forested regions (>40% tree cover; such as the Congo and Amazon Basins) are masked because soil moisture and VOD estimates are less certain from radiative transfer limitations in dense canopies (Feldman et al., 2018; Konings et al., 2017).

We compute the median $t_p$ over all drydowns within each 0.5° x 0.5° pixel. The $t_p$ probability mass function within a given pixel typically has a mixed distribution with many zeros, resembling a zero-inflated poisson distribution. The median $t_p$ is chosen to describe this distribution because it not only provides a typical timescale of VOD increase, but also indicates whether or not the majority of pulses resulted in consecutive, multi-day VOD increases (as opposed to the mean, which can be greater than zero even if a majority of pulses resulted in a $t_p$ of zero). Several tests are performed to determine the effects of SMAP's irregular, above daily sampling period, the algorithm, and measurement noise on $t_p$ estimates for a given pulse (see Section 2.4).

The $t_p$ definition evaluates continuous post rainfall VOD increases and potentially neglects the duration of plant water content increases during the period of soil moisture increase (between the observations before the drydown beginning and at the drydown beginning). We do not attempt to estimate the duration of VOD increase during the soil moisture increase period because it is not possible to resolve when plant water content increases initiated due to the 1-3 day satellite sampling frequency. Instead, the VOD behaviour preceding the drydown is categorically evaluated by determining the frequency of plant water content increases during the rain pulse. This allows evaluations of $t_p$ of zero which can result from either a rapid rehydration response during the rain storm (on the order of hours) or no rehydration response throughout the pulse (no VOD increase).

For each soil moisture pulse within a pixel, $t_p$ is estimated along with the LAI change from beginning to end of the drydown (ΔLAI), antecedent surface soil moisture (soil moisture value before drydown beginning), soil moisture pulse magnitude (difference between initially pulsed and antecedent surface soil moisture), and antecedent VOD. Antecedent is defined here as the observation just preceding the peak soil moisture observation beginning the drydown. Each variable is binned into rapid VOD response ($t_p$=0), short VOD increase (1≤$t_p$≤3 days), and long VOD increase ($t_p$ >3 days) groups because they provide partitions consistent with the satellite sampling and because uncertainty analyses reveal that while a $t_p$ estimate for a given drydown is uncertain, there is more confidence in whether it exists within a given bin (see Section 3.4). The groups of three different $t_p$ lengths are then compared for each respective metric of ΔLAI, antecedent surface soil moisture, soil moisture pulse magnitude, and antecedent VOD. Due to non-normality of groups based on Jarque-Bera normality tests (Jarque and Bera, 1980), Kruskal-Wallis non-parametric tests are performed to determine significance of difference in medians between the $t_p$ groups for each respective metric. Also, correlation coefficients are computed between $t_p$ and ΔLAI, antecedent moisture, and pulse magnitude to augment the categorical analyses.

The seasonal timing of rapid, short, and long $t_p$ values are assessed relative to peak seasonal moisture, or the proximity to the wet season. The peak seasonal soil moisture is determined by smoothing the soil moisture times series using a 90-day moving average window. This only provides a zeroth-order seasonal moisture peak approximation as many locations have intermittent rainfall or bimodal precipitation distributions.

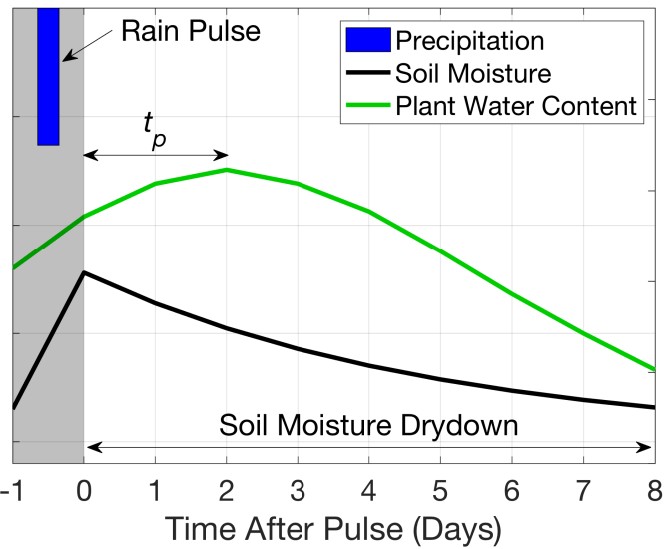

Fig. 1. Schematic definitions of rain pulse, soil moisture drydown period, and time to peak plant water content ($t_p$).

**2.4 Satellite plant water content response uncertainty analysis**

     Several tests were conducted to evaluate the robustness of $t_p$ estimates given uncertainties due to a 1-3 day satellite sampling frequency, the soil moisture-VOD retrieval algorithm, and random instrument noise on order of that of SMAP radiometer (Piepmeier et al., 2017). A stochastic rainfall generator was used to simulate soil moisture and consequent drydowns. A range of "true" VOD behaviour was considered such as perfect correlation with soil moisture (true $t_p$ of zero) and multi-day VOD increases during drying (true $t_p$ greater than zero). Analyses were conducted directly on these simulated time series, including converting these time series to TB measurements for implementation in the algorithm and comparing the original "true" VOD time series to the algorithm-estimated VOD time series as in Zwieback et al. (2019). For tests with the 1-3 day satellite satellite sampling frequency, the effect of randomly removing observations every 1-2 days on $t_p$ was assessed. To test the effect of the soil moisture-VOD retrieval algorithm on $t_p$, $t_p$ was estimated after inputting true TB measurements into the retrieval algorithm. Finally, to assess the effect of instrument noise on $t_p$ estimates, this aforementioned process was repeated by adding normally distributed random error to TB measurements.

**2.5 Plant Hydraulic Model Simulations**

     To investigate the underlying mechanisms that alter plant rehydration timescales, we evaluate plant hydraulic storage timescales under varying conditions after a surface soil moisture pulse using a plant hydraulic model. We specifically choose a one-dimensional soil-plant-atmosphere continuum (SPAC) model, assessed in previous studies (Carlson and Lynn, 1991; Hartzell et al., 2017; Lhomme et al., 2001; Zhuang et al., 2014). Note that assimilating satellite VOD into a SPAC model is beyond the scope of this study and is hindered by the large number of unknown plant hydraulic parameters at global scales. SPAC simulations are repeated and randomized using a Monte Carlo approach, drawing from parameter distributions based on previous field measurements. More details can be found about the SPAC model in the SI.

**3. Results**

**3.1 Global plant water content characteristic responses and timescales**

The VOD data shows that more arid regions, with lower annual rainfall and tree cover (Fig. 2B and 2C), exhibit multi-
day vegetation water content increases following moisture pulses ($t_p \geq 1$ day, blue regions in Fig. 2A). That is, after soil moisture
increases following a storm, vegetation water content increases for multiple days even while surface soil moisture begins to
dry. Furthermore, in regions with $t_p \geq 1$ day, VOD typically begins increasing during the rain pulse period instead of with a lag
after soil moisture drying begins (occurs in 77% of the pixels). Aggregated example time series of this nonzero $t_p$ behaviour
can be seen in drylands in the Sahel and Southwest United States (Figs. 3A and 3B). In the regions with multi-day VOD
increases, the spatial median $t_p$ is two days. Note that various responses are spatially aggregated together to produce the post-
rainfall responses in Fig. 3. In subsequent sections, we evaluate and partition the mechanisms underlying these multi-day plant
water content increases primarily in drylands (blue regions in Fig. 2A).

      By contrast, more humid ecosystems with more woody plant coverage typically do not exhibit multi-day plant water
content increases ($t_p=0$; Fig. 2). They instead exhibit water loss following the pulse during soil drying (see average behaviour
illustrated in Figs. 3C and 3D). In 83% of regions with $t_p$ of zero (red regions in Fig. 2A), the plant drying responses are
typically preceded by an initial VOD increase, showing rapid water uptake during the storm period (Fig. S2). In contrast, a
minority of these regions typically show no VOD increases, suggesting plant water content continuously dries throughout the
pulse with no discernable hydraulic response (Fig. S2). We do not investigate regions with median $t_p$ of zero further here
because their exact sub-daily timescales are unresolvable, but within expectations (see Discussion).

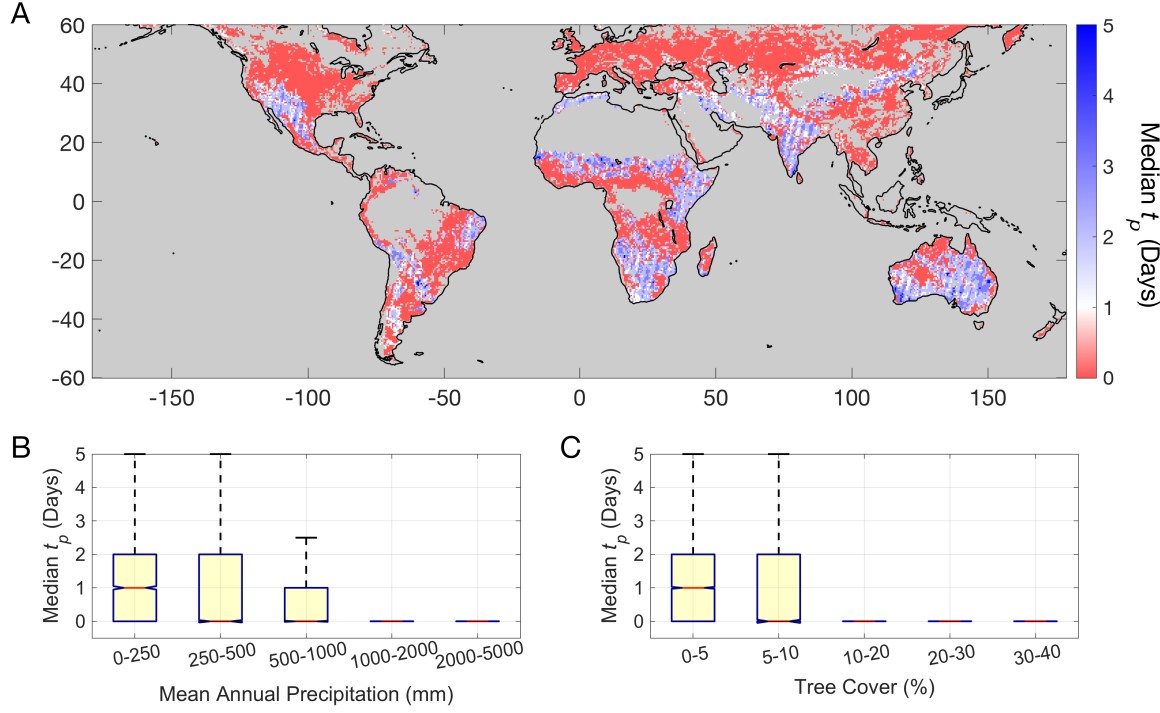


Fig. 2. Median time to peak plant water content ($t_p$) after soil moisture pulse. (A) Median $t_p$ global distribution. Median $t_p$
binned as a function of (B) mean annual precipitation and (C) tree cover. Mostly bare surfaces with low vegetation density are
masked. Densely forested areas (tree cover > 40%) are masked due to limitations in VOD estimation for dense canopies.

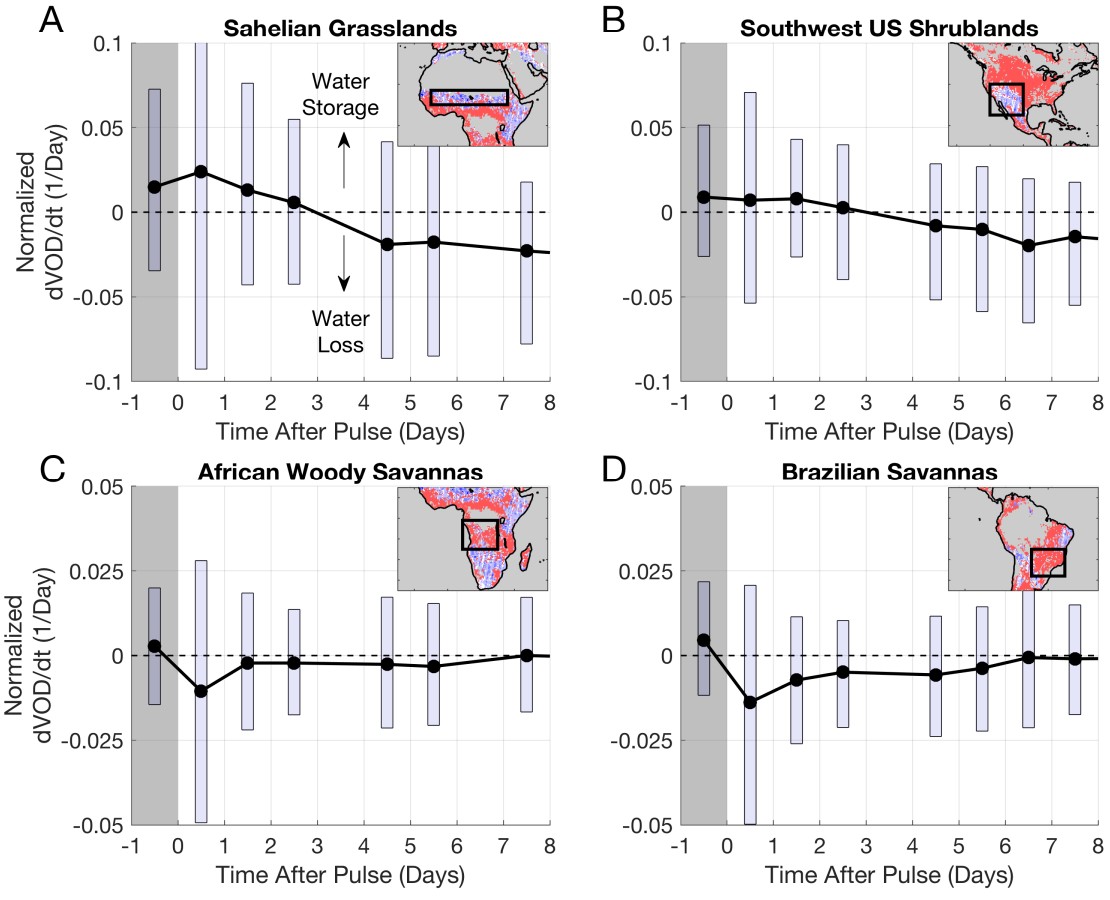


Fig. 3. VOD rate of change distribution on a given day after the pulse for regions outlined in the insets. Boxes delineate the interquartile range for each day. dVOD/dt is normalized by dividing by VOD time mean for a given pixel for consistent comparison across regions. dVOD/dt is reported as the average change rate over a given day (for example, from day 0 to day 1). All pixels with the noted dominant land cover (>75% IGBP land cover type) are used within the boxed region in the inset

to create the distributions for each respective day after the pulse. Gray shading indicates the pulse period when soil moisture is increasing (Fig. 1). At time greater than zero, soil moisture is drying (drydown event; see Fig. 1). Behaviour extends beyond a week in many cases, but only eight days following the pulse are shown here. Note that top and bottom panels have different y-axis limits.

**3.2 Growth influence on plant water content increase timescales**

A positive correlation between LAI rates of change and plant water content increase timescales is found in 72% of African pixels with median $t_p \geq 1$ ($p<0.05$). Therefore, longer $t_p$ are associated with increasing biomass within a given pixel (Fig. 4). Calculating the LAI rates of change for the rapid VOD response ($t_p=0$), short VOD increase ($1 \leq t_p \leq 3$), and long VOD increase ($t_p>3$) groups reveals that growth tends to occur alongside plant water content increases longer than three days (Figs.

5A, 5C, and 5D). These longer plant water content uptake timescales average 7 days and continue beyond a week 40% of the time. This growth influence means that rehydration alone cannot explain longer plant water content increase durations. Note that VOD increases during growth still demonstrate increased aboveground plant water content because more aboveground

biomass requires water uptake to hydrate a greater volumetric plant storage capacity. There are some pixels that show declining biomass during longer $t_p$ (Fig. 5D). We attribute these cases to detection of longer $t_p$ during senescence in regions where senescence of leaf area is differentially more rapid than growth. Ultimately, we interpret overall spatial patterns and avoid interpreting individual pixels, acknowledging noisy $t_p$ estimates in some cases (see Section 3.4).

In general, growth does not influence shorter plant water content increase timescales; LAI is often decreasing when $t_p$ is 1-3 days (Fig. 5A). Therefore, plant water content increases over less than three days are mostly due to rehydration. Furthermore, when VOD increases do not extend beyond a day ($t_p = 0$), growth is also less frequently occurring.

The reoccurrence of growth influenced, multi-day VOD increases consistently following soil moisture pulses means that rainfall intermittently triggers growth throughout a year. $t_p > 3$ are linked to pulse driven growth because they coincide with increasing daily LAI (Fig. 5), consistently co-occur with a soil moisture pulse, and are separated from seasonal growth patterns. Our seasonal detrending of VOD isolates these pulsed plant growth responses from seasonal growth cycles. These isolated sub-weekly VOD responses closely link to the timing of moisture pulses suggesting a cause-effect of rain pulse followed by plant water content response.

Although this daily LAI dataset is limited to Africa only, Africa contains one third of the world's regions with median $t_p \geq 1$ day (blue regions in Fig. 2A) and we expect similar results for the rest of the globe. Note that these results are not sensitive to the three-day threshold choice between long and short VOD increase groups; they are nearly identical if choosing a threshold of two, four, or five days. Furthermore, results repeated with FAPAR are qualitatively the same (Fig. S1; see Section 2.1).

On average, the short and long VOD increase bins occur approximately with equal frequency, both with seasonal variations (Fig. 5B). Longer duration VOD increases influenced by growth (Fig. 5A) appear to occur more frequently during times of the year when soil moisture is higher (Fig. 5B). In contrast, short VOD increases, associated more with rehydration, occur more often during drier times of the year (Fig. 5B). Furthermore, rapid rehydration responses occur 40-50% of the time throughout the year amongst the multi-day VOD increases.

LAI growth rates average 0.005 $m^2/m^2$ per day for these long VOD increases. On a mean percent change basis, this translates to a 15% LAI increase on average over the course of a week after a pulse. Note that LAI may not detect additional branch/stem biomass growth that VOD may detect. Ultimately, we are more interested in qualitatively increasing trends in LAI rather than the magnitudes of LAI rates of change which are less certain.

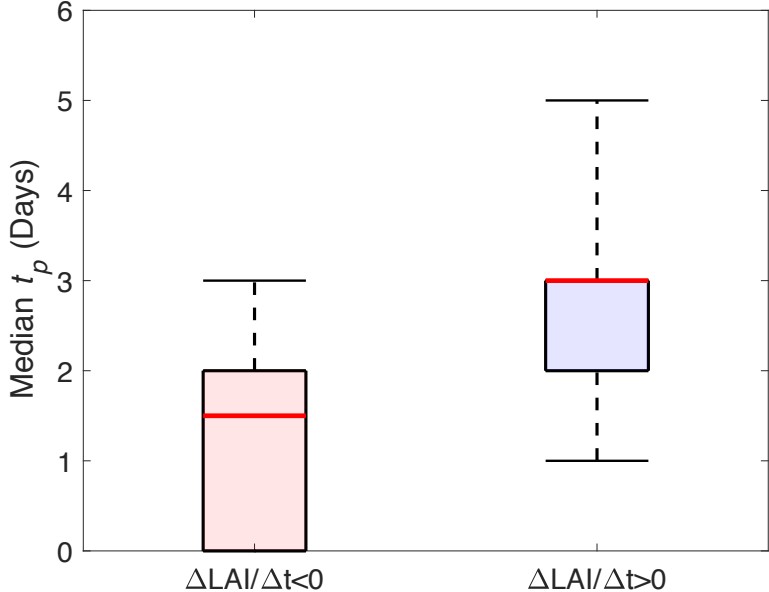


Fig. 4. Relationship of plant water content increase timescales with biomass changes in African regions with median $t_p \geq 1$ day. Growth increases the timescale of plant water content. Mann-Whitney U tests indicate that the medians of the two bins are significantly different ($p<0.05$).

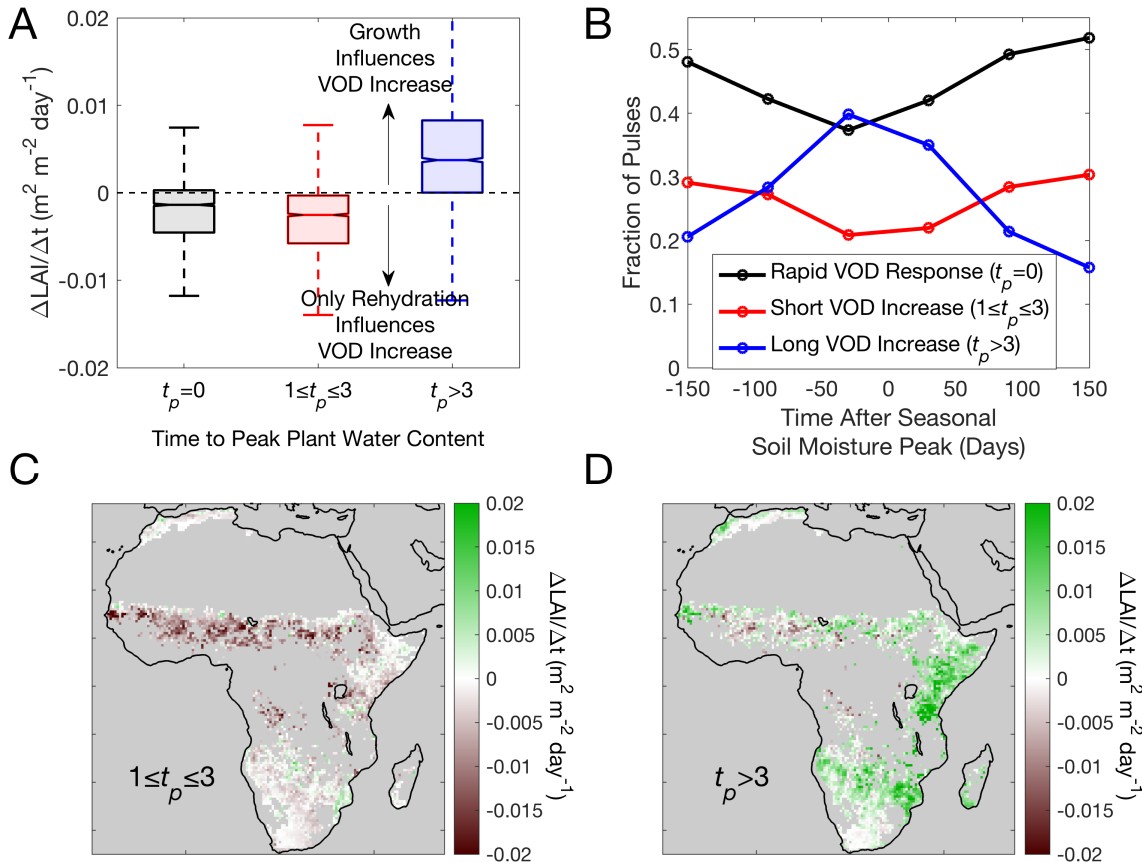

Fig. 5. Timescale of plant water content increases relation to biomass changes and seasonality in African regions with median $t_p \geq 1$ day. Growth influences the plant water uptake timescale when $\Delta LAI/\Delta t > 0$. By contrast, only rehydration contributes to plant water content increases when $\Delta LAI/\Delta t < 0$. Only intermittent variability in VOD is used to produce $t_p$, removing seasonal confounding connections with LAI (see text and SI). (A) Mean change in LAI per day over length of pulse period binned into rapid responses ($t_p = 0$), short VOD increases ($1 \leq t_p \leq 3$ days), and long VOD increases ($t_p > 3$ days). A Kruskal-Wallis test indicates group medians are all significantly different ($p \ll 0.01$; $\chi^2 = 2{,}576$, $\upsilon = 2$). Pairwise Mann-Whitney U tests confirm that all pairs are significantly different ($p < 0.05$). (B) Seasonality of short and long VOD increase occurrences with respect to seasonal soil moisture peak. Positive and negative time indicates occurrence after and before the soil moisture seasonal peak, respectively. Plotted values are spatial medians in 60-day sized bins. Sample size in each bin (in a given pixel) is over 100, though pulses tend to be more frequent nearer to seasonal soil moisture peak. (C) Spatial distribution of median $\Delta LAI/\Delta t$ for short VOD increases as binned in A. (D) Spatial distribution of median $\Delta LAI/\Delta t$ for long VOD increases as binned in A.

### 3.3 Pulse condition influence on plant water content increase timescales

Variations in VOD increase timescales across space and time likely occur as a result of differences in vegetation traits, edaphic and topographic properties affecting soil moisture infiltration, and climatic properties. While an evaluation of

all of these factors is beyond the scope of this manuscript, we focus here on climatic drivers. To evaluate the climatic drivers of VOD increase timescales in regions with median $t_p \geq 1$ day (blue regions in Fig. 2A), we assess how $t_p$ relates to rain pulse conditions: antecedent surface soil moisture, soil moisture pulse magnitude, and antecedent VOD. Growth-influenced VOD increases of longer duration are associated with initially wetter surface soil (Fig. 6A) as well as with larger pulse magnitudes (Fig. 6B). This suggests that the surface must be sufficiently wet initially and a large enough pulse must occur to elicit a growth

response. Conversely, shorter duration VOD increases associated primarily with rehydration frequently occur under drier initial soil conditions with smaller rewetting pulses (Fig. 6). This is consistent with short increase durations becoming more prevalent during drier periods and long increase durations becoming more prevalent in wet periods (Fig. 5B). Note that while these results are shown globally, they are nearly identical when calculated for only Africa (not shown), and therefore can be consistently compared with the growth assessment results and timescale bins (Sect. 3.2; Fig. 5).

330          In assessing what differentiates rapid responses ($t_p$ = 0 days) and short VOD increases ($t_p$ = 1-3 days) that appear driven by only rehydration, we find short VOD increases have slightly larger pulse magnitudes (Fig. 6B) and drier antecedent soil moisture than rapid responses (Fig. 6A). Also, drier initial plant water status for short VOD increases (Fig. 6C) independently suggests a slightly drier root zone initially than for rapid responses (Fig. S13). Note that mean differences are small between these metrics, even though they show statistical significance (likely effect of large sample size deflating p

values). Nevertheless, cases of vegetation water content increase on the order of 1-3 days, due primarily to rehydration, occur under dry soil conditions with small-to-moderate rewetting pulses.

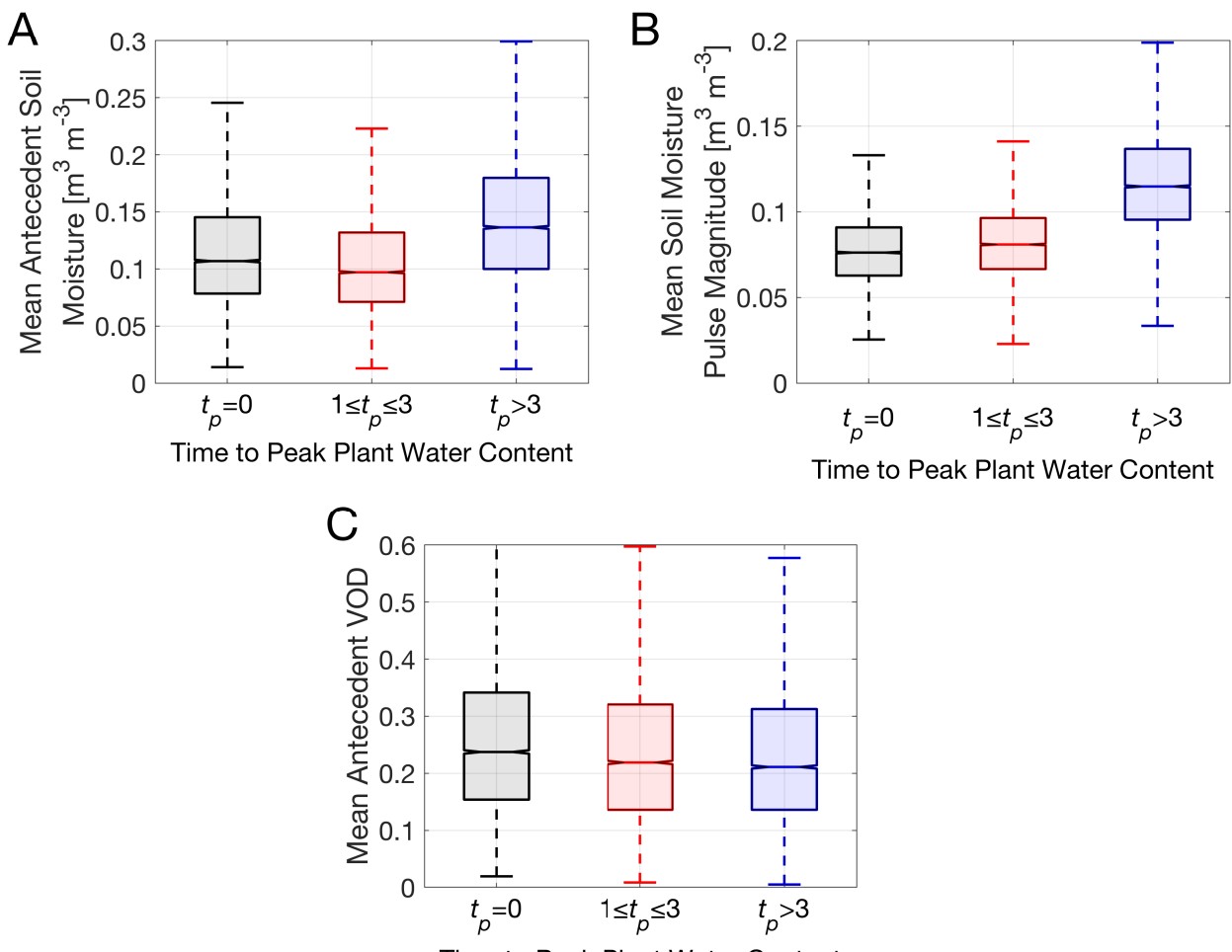

Fig. 6. Global spatial distribution of pulse conditions binned as function of rapid VOD response ($t_p =0$), short VOD increases ($t_p$ =1-3 days), and long VOD increases ($t_p >3$ days) in regions with median $t_p \geq 1$ day. Kruskal-Wallis (KW) tests indicate all group medians are significantly different within each panel and pairwise Mann-Whitney U tests confirm that all possible combinations of differences in group medians across A, B, and C are significantly different ($p<0.05$). (A) Antecedent surface soil moisture (KW test $p<<0.01$; $\chi^2=2,200$, $\upsilon=2$). 77% of pixels have significantly positive linear relationships with $t_p$ ($p<0.05$). (B) Surface soil moisture pulse magnitude (KW test $p<<0.01$; $\chi^2=7,819$, $\upsilon=2$). 85% of pixels have significantly positive linear relationships with $t_p$ ($p<0.05$). (C) Antecedent VOD (KW test $p<<0.01$; $\chi^2=163$, $\upsilon=2$). 81% of pixels have significantly negative linear relationships with $t_p$ ($p<0.05$).

### 3.4 Satellite plant water content response uncertainty analysis

Satellite $t_p$ estimates appear robust with effects of satellite sampling frequency, algorithmic estimation error, and measurement noise increasing $t_p$ variance, but not introducing discernable biases. The SMAP sampling period of 1-3 days results in greater variance but no mean biases for $t_p$ estimates below the Nyquist frequency of 4-6 days (Figs. S4 and S5). One can combine low frequency microwave measurements from similar satellites (Kerr et al., 2010) to increase the sampling frequency and reduce uncertainty in $t_p$ estimates here. This is not attempted due to complications in combining the datasets.

The MT-DCA algorithm used here reduces sensitivity to noise within the simultaneous soil moisture-VOD estimation (Konings et al., 2016, 2015; Zwieback et al., 2019). We found that use of a traditional algorithm biases $t_p$ towards zero (Fig. S7) because its greater sensitivity to noise will tend to spuriously induce positive correlation between soil moisture and VOD within the estimation procedure (Konings et al., 2016). Therefore, increases in VOD during soil drying and thus positive $t_p$ are not a result of algorithmic artifacts from the MT-DCA algorithm used here (Feldman et al., 2018). It is also unlikely that

algorithmic noise is driving spatial patterns as both algorithms produce the same $t_p$ spatial patterns. Note that the MT-DCA algorithm can slightly, artificially increase $t_p$, though measurement noise may cancel this effect (Fig. S4). Finally, measurement noise primarily increases the variance of $t_p$ (Fig. S4).

         Ultimately, while identifying precise $t_p$ values for a given drydown may be hindered by these sources of uncertainty, median $t_p$ for a pixel are likely not biased and more confidence is exhibited in whether $t_p$ is zero or non-zero (Fig. S6). This

uncertainty analysis provides confidence in the global patterns of median $t_p$ and results based on binned $t_p$ where zero, short, and long $t_p$ can be confidently partitioned.

## 4. Discussion

### 4.1 Plant water uptake timescale variation across climates

We observe a continuum of plant water uptake timescales from humid to dryland environments, with mainly drylands showing frequent multi-day plant water content increases after rainfall before water loss occurs (Fig. 2). Given that plant hydraulic capacitance increases at least three orders of magnitude from grasses in drylands to trees in humid regions (Carlson and Lynn, 1991; Hunt Jr et al., 1991), one might expect, if at all, occurrence of multi-day responses in wooded regions. However, humid, wooded regions broadly exhibit peak plant water content during rather than after the storm event, before soil

drying begins (Figs. 2 and S2). Plant water loss occurs thereafter (Figs. 3C and 3D), likely due to simultaneous soil and plant drying where plant rehydration becomes progressively restricted with drying soil (Feldman et al., 2020b). The initial VOD increase can be due to plant water uptake where pre-dawn water potential approaches equilibrium with soil moisture and/or due to plant interception of rainfall droplets. In some cases, no discernible VOD increase occurs before or after the pulse, which may indicate sufficiently well-watered conditions (Fig. S2). Even in drylands, pulse water utilization for plant

rehydration decreases if the plant-soil system is initially sufficiently wet (Ehleringer et al., 1991; Gebauer et al., 2002; Ignace et al., 2007). Nevertheless, due to the 1-3-day satellite sampling, we are unable to resolve more specific plant water content timescales and underlying mechanisms for these well-watered, wooded regions.

         The consistent trend of multi-day plant water content increases, which are found broadly across dry regions (Fig. 2), is unexpected, at least in the context of nominal RC time constants (plant water uptake and storage timescales). Field-based

estimates of plant water uptake timescales (via RC plant hydraulic time constants) typically do not exceed a day, regardless of species (Huang et al., 2017; Nobel and Jordan, 1983; Phillips et al., 1997, 2004; Ward et al., 2013). This is in part because plant capacitance and resistance tend to trade off with changes in plant architecture and moisture conditions (i.e., capacitance increases and resistance decreases generally from grass to tree species) (Hunt Jr et al., 1991; Phillips et al., 1997; Richards et al., 2014; Ward et al., 2013). We find both the influence of growth and slow plant rehydration contribute to these observed

multi-day VOD increases. We discuss these growth and plant rehydration mechanisms observed in drylands further below.

### 4.2 Growth impact on dryland plant water uptake timescales

         As is evident in independent satellite LAI observations, growth increases the duration of plant water content increases (Fig. 4), and appears to occur primarily for plant water content increases of more than three days in dryland regions (Fig. 5).

These week-long consecutive plant water content increases occur when the soil is initially wetter and pulses are larger (Fig. 6). These results are based on 1-2 week increasing trends in LAI coinciding with VOD increases of more than three days. Confidence is exhibited in these sub-monthly LAI trends because of SEVIRI's ability to resolve the seasonal growth stages during the wet season, lower LAI uncertainty in Africa's biomes with herbaceous vegetation, and SEVIRI's filtering of LAI

noise. Therefore, plant rehydration alone cannot explain these longer duration VOD increases. We further suspect rehydration is rapid under these well-watered conditions. While pulsed-growth is expected to occur with a lag of 1-5 days (Ogle and Reynolds, 2004), these lags may be obscured in the sampling of VOD and initial VOD increases due to rehydration. Furthermore, these pulsed plant water content increases due to growth may continue for longer than detected here (beyond two weeks). However, continued water loss and VOD decreases through transpiration may eventually dominate over VOD increases due to growth, curtailing the peak VOD (resulting in behaviour like that shown schematically in Fig. 1). VOD ultimately shows sub-weekly growth temporal dynamics beyond those resolved from optical instruments.

These results indicate that large soil moisture pulses on initially wetter soils trigger dryland vegetation growth responses after storm events, as hypothesized under the pulse-reserve paradigm (Collins et al., 2014; Noy-Meir, 1973). This weekly variability, at least in part, drives seasonal growth in these locations (Reynolds et al., 1999) where the seasonal growth cycles appear to be made up of sub-weekly intermittent growth dynamics as modeled in Ogle and Reynolds, (2004). The growth occurrences under wetter conditions are expected given that cell turgor must be high for cell expansion and rapid growth to occur (Kramer and Boyer, 1995). Furthermore, a recent study showed that larger pulses during the growing season resulted in 1-2 weeks of increasing leaf and stalk density in a semi-arid grassland, consistent with results here (Post and Knapp, 2019). Additionally, larger pulses have previously been shown to elicit greater plant photosynthetic responses (Chen et al., 2009; Dougherty et al., 1996; Schwinning and Sala, 2004). In a similar study, these longer satellite-based plant water uptake responses coincided with larger and longer carbon uptake responses at dryland flux tower sites following larger moisture pulses on initially wet soils (Feldman et al., 2020a). Therefore, detection of pulse-triggered growth on timescales of drydowns here is consistent with previous results, although it is the first to show how widespread the pulse-triggered growth dynamics are in drylands. Additionally, the seasonal occurrence of growth-driven, longer $t_p$ (Fig. 5B) supports that pulses will trigger growth primarily in the season when species are phenologically active and able to invest in aboveground biomass (Post and Knapp, 2019; Reynolds et al., 1999; Schwinning and Sala, 2004).

**4.3 Slow dryland plant rehydration mechanisms**

Over half of the moisture pulses, primarily in global drylands, result in multi-day satellite-observed plant water content increases (Fig. 2). These multi-day VOD increases are often only due to rehydration, especially the shorter VOD increases (1-3 day) following small-to-moderate pulses on initially dry soils (Figs. 5 and 6). They can occur even when biomass is decreasing (Fig. 5C; such as leaf off), where the relative water content increases are larger than what the VOD increase signal suggests. For dryland ecosystems that include grass and shrub species with isolated forests, multi-day rehydration is generally unexpected with nominal RC time constants on the order of an hour (Carlson and Lynn, 1991; Hunt Jr et al., 1991). However, previous field studies often show 1-4-day rehydration of grasses and shrubs upon rewetting following dry conditions, especially in the southwestern United States, where multi-day VOD increases are observed (Briones et al., 1998; Fravolini et al., 2005; Huxman et al., 2004; Ignace et al., 2007; West et al., 2007).

To better understand the physiological drivers of multi-day rewetting, we assessed the potential hydrologic and physiological mechanisms driving slow rehydration using a plant hydraulic (SPAC) model and parameters within known bounds for semi-arid species (Figs. S8 to S14 and Table S1). We find that the sufficient conditions for multi-day plant rehydration determined here include initially high soil-plant resistances decreasing over multiple days following a storm. These time-varying resistances can occur either in the soil, plant, or both (Figs. 7, S9, and S11). The possibility of multi-day rehydration due to these conditions suggests that RC timescales can greatly deviate from nominal conditions (Scholz et al., 2011), especially under drought scenarios where resistances are both higher and changing.

After uncoupling effects of soil and plant resistances in the SPAC model, we suspect that multi-day rehydration as seen by VOD is dominated by plant resistance limitations rather than soil resistance limitations. This is because high soil resistances reduce infiltration rates and result in a phase-lagged delay in plant rehydration (Fig. 7B), which is not observed in the satellite VOD behaviour here. In the slow rehydration cases ($t_p$ = 1-3 days), VOD increases begin immediately during the

storm and not with a phase lagged delay (Fig. S3). This behaviour more closely resembles slow plant rehydration dominated by plant resistance limitations rather than those dominated by soil resistance limitations. For example, 1-3-day uptake
timescales based on satellite VOD observations appear like that in Figs. 3A and 3B, which more closely resembles SPAC model simulations in Fig. 7A than in Fig. 7B. Note that both conditions may be present within a coarse-resolution pixel because the pixel spatially averages plant water content behaviour over the landscape. As a result, a combination of behaviours like those in Fig. 7 aggregate into the spatially averaged behaviour, like that shown in Figs. 3A and 3B. Therefore, while plant resistance limitations may dominate most landscapes that show 1-3-day VOD increases based on the above discussion, slow
infiltration responses may still be spatially prevalent, with a potential dependence on sub-pixel antecedent moisture variability.

The initially high, decreasing resistances, as determined from the SPAC model and likely influencing landscape scale plant water content behaviour, are likely due to drought recovery of soil-root interface and xylem architecture. Initially high, decreasing plant resistances have been observed in the field, where after rewetting of dry soil conditions, soil-root interface and xylem resistances can decrease by one to three orders of magnitude over a few days (Carminati et al., 2017; North and
Nobel, 1995; Trifilò et al., 2004; West et al., 2007). Under prolonged dry conditions, a disconnect between soil and root interface can occur, and after rewetting, the soil-root and radial root hydraulic conductivity progressively increase (Carminati et al., 2009; North and Nobel, 1997). Similarly, xylem cavitation and embolism from drying lead to increased xylem resistance that can regain conductance and refill after rewetting (Martorell et al., 2014), though noting controversies with existence of xylem repair and refilling (Charrier et al., 2016; Lamarque et al., 2018; Venturas et al., 2017). Recent evidence suggests that
whole root resistance (i.e., soil-root interface, radial) rather than xylem resistance (from cavitation) dominate the whole plant resistance during these drying and rewetting cycles (Rodriguez-Dominguez and Brodribb, 2020). Finally, fine root growth can occur after rewetting which can contribute to decreasing root resistances, though these effects may occur over longer, weekly scales (Eissenstat et al., 1999).

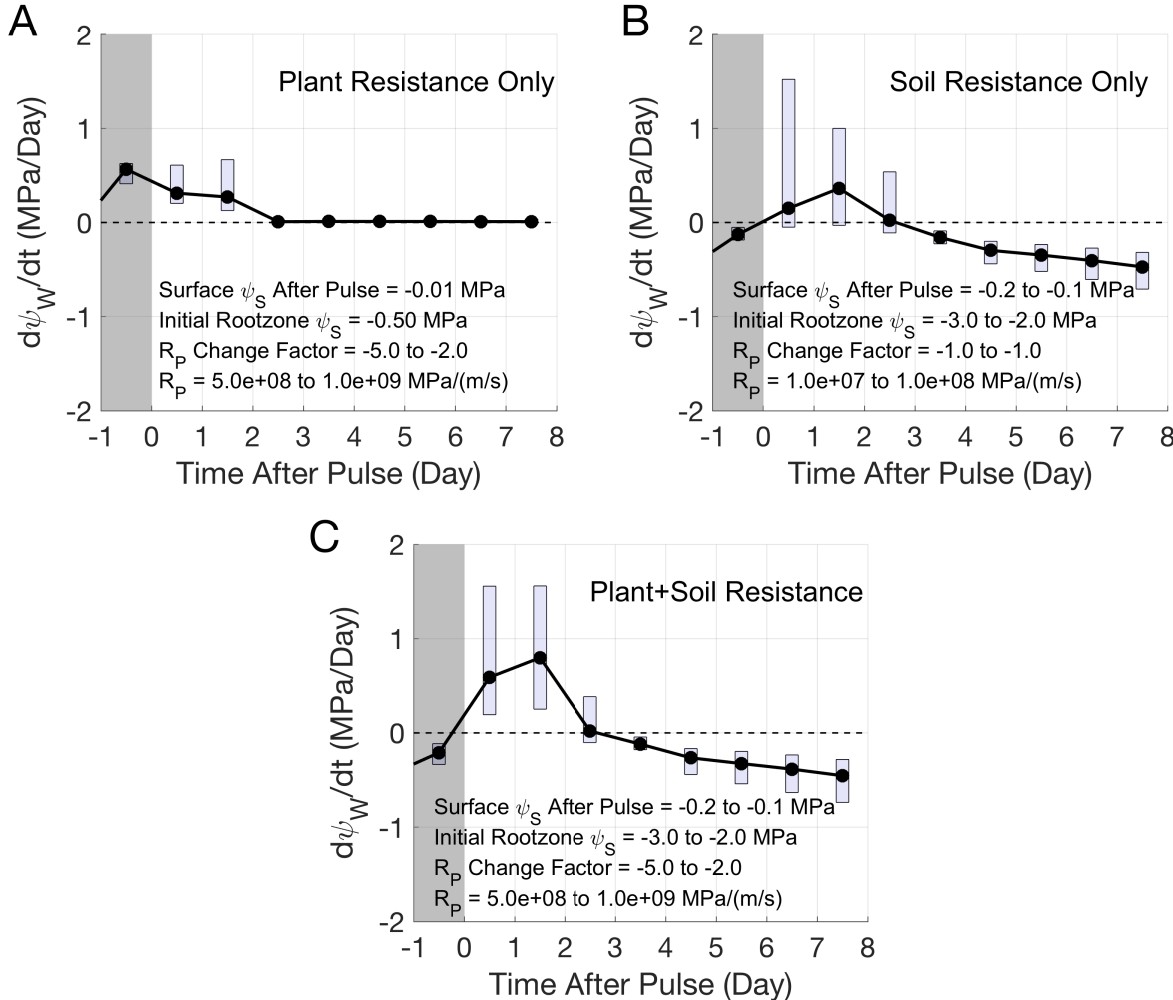

Fig. 7. SPAC model simulations of determined sufficient conditions driving slow rehydration (see text and SI) for semi-arid grass and shrub species. Rate of change in predawn water potential ($\psi_w$) of all plant water stores on a given day following a pulse where $d\psi_w/dt > 0$ indicates rehydration. Same format and conventions as Fig. 3. Parameter bounds determined to drive each slow rehydration scenario are shown in each panel. (A) Plant limitation only where plant resistance ($R_P$) is initially high and decreases. (B) Soil limitation only where rootzone soil moisture is initially dry and pulses are small to moderate, causing slow infiltration. (C) Both plant and soil limitations superposed from A and B. Parameter ranges common amongst all simulations: rooting depth = 0.3 to 0.7 m, VPD = 1 to 5 kPa, wind = 1 to 8 m/s, C = $10^{-6}$ to $10^{-5}$ m/MPa, and $R_S$ impairment factor = -10 to -1. See SI for more information on the SPAC model and simulations.

## 5. Conclusions

The globally-observed timescales of plant water content responses to moisture pulses here reveal a climate gradient of ecosystem-scale vegetation pulse water usage. The vegetation water content of more wooded, humid regions appears to respond rapidly to rain pulses, likely with rehydration responses occurring in less than a day (due to predawn equilibrium). By contrast, drier ecosystems more often show multi-day plant water uptake responses following moisture pulses with the timescale of the response indicative of underlying mechanisms. Specifically, longer plant water content increases are linked

to growth and follow larger pulses on wetter surfaces. Therefore, dryland vegetation intermittently upregulates and grows after individual rainfall events, demonstrating spatially-extensive evidence for the pulse-reserve hypothesis. Specifically, we show that there is a component of growth linked directly to individual rainfall events in addition to any continuous seasonal growth (Noy-Meir, 1973). Additionally, shorter plant water content increases are indicative of slow plant rehydration responses and

are linked here to hydraulic recovery from initially dry conditions. The slow rehydration responses indicate that plant water uptake timescales can frequently deviate from nominal RC time constants with greatly increased resistances under dry conditions, as observed previously in field experiments and demonstrated here using a SPAC model.

Our results also indicate that SMAP satellite vegetation optical depth observations hold biophysical information at sub-weekly timescales. Namely, they show patterns of rehydration, growth responses, and rain pulse dependencies consistent

with that seen in field studies. These satellite-based plant water content responses were also shown to have similar response signatures to carbon uptake responses at dryland field sites (Feldman et al., 2020a). This merits investigation of sub-monthly ecological processes using these 1-3 day sampled satellite microwave observations, which so far have been primarily used for seasonal and interannual VOD variability investigations (Brandt et al., 2018; Jones et al., 2014; Tian et al., 2018).

We demonstrate that global dryland ecosystems exhibit a high sensitivity to the characteristics of individual moisture

pulses. Therefore, expected shifts in rainfall frequency and intensity may influence arid to semi-arid vegetation hydraulic and growth processes, presenting potential feedbacks on biogeochemical cycles and changes in plant community composition (Giorgi et al., 2019; Knapp et al., 2002). These dry ecosystems cover 40% of the land surface, store significant amounts of carbon (Beer et al., 2010; Collins et al., 2014), regulate atmospheric carbon interannual variability (Ahlström et al., 2015; Poulter et al., 2014), and are projected to expand (Huang et al., 2016). Therefore, it is key to characterize the vegetation

responses to rainfall events – including their timescales - in these environments in the context of predicting future climate.

**Acknowledgements**

Massachusetts Institute of Technology contributors acknowledge funding from NASA in the form of a sponsored research grant (Subcontract No. 1510842). A.G.K. was supported by NASA Terrestrial Ecology (grant number 80NSSC18K0715). P.G.

and A.G.K. acknowledge NOAA award NA17OAR4310127. The authors thank Missy Holbrook, Tony Rockwell, Anju Manandhar, and Jess Gersony of the Holbrook Plant Physiology Laboratory at Harvard University for many insightful discussions. The authors also thank two anonymous reviewers for their insightful comments.

**Author Contribution**

P.G. and A.F.F. conceived the study. D.E. led the project. A.F.F. conducted the analysis and wrote the manuscript. D.J.S.G., A.G.K., P.G., and D.E. contributed interpretations and numerous revisions to all versions of the manuscript, analysis, and figures.

**Data Accessibility**

SMAP L1C brightness temperature used to retrieve soil moisture are available from the National Snow and Ice Data Center (NSIDC) (https://nsidc.org/data/ SPL1CTB_E). LandSAF leaf area index is available from EUMETSAT (https:// landsaf.ipma.pt). Generated maps are available at https://github.com/afeld24/VOD_Timescales.

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
