# Peer review of "Patterns of plant rehydration and growth following pulses of soil moisture availability"

_Biogeosciences, 2020_

## Referee Comment (RC1) · Anonymous Referee #1 · 30 Oct 2020

Summary This study assesses the temporal pattern of soil moisture and plant water content derived from SMAP to understand plant responses to precipitation pulses, which have shown significant lags in field studies. The comprehensive spatial scale of this analysis is impressive and timely. Importantly, the authors use a daily LAI product for Africa to account for the combined growth and rehydration signal inherent in VOD interpretation. I have some concerns about the correct use of Kruskal-Wallis tests and found the comparison between spatially average temporal VOD trends and results of a 1D SPAC model somewhat spurious. Major comments Ln 282 - 287: This section is throwing me off because I cannot easily distinguish between rapid responses and short VOD increases; they sound the same to me. Adding the actual tp lengths would help. More importantly, the boxplots of Fig. 5 do not convey distinctions based on

the KW tests. Is there some kind of non-parametric post-hoc test, equivalent to Tukey HSD, which can be used to put labels on the boxplots? As far as I am aware, the KW tests only indicate that at least one group is significantly different but does not indicate which ones. This may require pairwise Mann-Whitney tests. I don't think the spatially averaged time series shown in Fig. 3A and B are comparable to a 1D SPAC model result in Fig. 6A and B. Rather, both kinds of modeled drydown patterns might be found in the actual data if not aggregated spatially. The relatively occurrence of immediate versus lagged rehydration may additionally depend on antecedent soil moisture conditions. Minor comments Ln 160: Does this refer to the empirical probability mass function? Probably more clear/accurate to referr to as the histogram and described as a zero-inflated Poisson. Ln 176-184: This section could use greater clarity. Make clear that non-parametric ANOVAs are to compare the covariates of ∆LAI, antecedent soil moisture, antecedent VOD, etc. between the three groups of VOD pulse responses. Ln 231: "... aboveground biomass requires water uptake ..." Ln 238: "co-occur" Ln 238-241: I suggest "Seasonal detrending of VOD isolates plant rehydration response to moisture pulses, which show multi-day increases and eventual decreases following moisture inputs." Current phrasing is difficult to understand. Section 3.3: Is this section also limited to Africa? Make clear at the outset, rather than in Ln 278-280. Otherwise, "growth-influenced VOD" is not supported. Ln 299-306: This uncertainty analysis should be at least mentioned in the Methods section and lead with the purpose of this approach before referring the reader to the supplement. Ln 326: Plant hydraulic capacitance, or something else? Ln 332: "... pulse, which may indicate sufficiently well-watered conditions..." Ln 336-337: This sentence may not be necessary. Is the emphasis on parallel soil moisture and VOD decreases following the pulse? What is "plant-storage water potential"? Ln 339: Redefine RC time constants again at the beginning of the Discussion Ln 380: Again, I suggest introducing the SPAC model and its purpose in the Methods section before referring the reader to the supplement. Ln 406-408: This seems in direct contradiction to the above citation of Kramer and Boyer 1995. There is also the component of fine root growth after soil rewetting. Is the SPAC

model able to represent the soil-root conductance or potential growth of new roots? Ln 409-414: Please clarify the thoughts presented here. Is it to related VOD to water potential as well as water content? This paragraph seems out of place in this section and is not well-prefaced by the introduction. Ln 433: "...demonstrating spatially-extensive evidence for..." Ln 440: Use commas in the list Ln 444: "show" is used twice. Perhaps "We demonstrate...ecosystems exhibit ..."

---

## Author Comment (AC1) · 9 Nov 2020

Reviewer 1: This study assesses the temporal pattern of soil moisture and plant water content derived from SMAP to understand plant responses to precipitation pulses, which have shown significant lags in field studies. The comprehensive spatial scale of this analysis is impressive and timely. Importantly, the authors use a daily LAI product for Africa to account for the combined growth and rehydration signal inherent in VOD interpretation. I have some concerns about the correct use of Kruskal-Wallis tests and found the comparison between spatially average temporal VOD trends and results of a 1D SPAC model somewhat spurious.

Authors: Thank you for your constructive comments on our study. We will address your

comments as discussed in the responses below.

Reviewer 1: Major comments Ln 282 - 287: This section is throwing me off because I cannot easily distinguish between rapid responses and short VOD increases; they sound the same to me. Adding the actual tp lengths would help.

Authors: We will add in parentheses in line 281 that rapid VOD responses have tp = 0 days and short VOD increases have tp of 1-3 days.

Reviewer 1: More importantly, the boxplots of Fig. 5 do not convey distinctions based on the KW tests. Is there some kind of non-parametric post-hoc test, equivalent to Tukey HSD, which can be used to put labels on the boxplots? As far as I am aware, the KW tests only indicate that at least one group is significantly different but does not indicate which ones. This may require pairwise Mann-Whitney tests.

Authors: It is correct that the KW test does not identify where the large differences in means are. We did all possible pairwise Mann-Whitney U tests between all variables and found statistically significant (p<0.05) differences for all tests. We wish to avoid adding labels to the boxplot. Therefore, we will note that pairwise Mann Whitney U tests show significant differences in the Fig. 5 caption.

Reviewer 1: I don't think the spatially averaged time series shown in Fig. 3A and B are comparable to a 1D SPAC model result in Fig. 6A and B. Rather, both kinds of modeled drydown patterns might be found in the actual data if not aggregated spatially. The relatively occurrence of immediate versus lagged rehydration may additionally depend on antecedent soil moisture conditions.

Authors: We acknowledge the differences in spatial scale of the results. The idea is that the mechanism that dominates spatially will influence the VOD signal the most, while the other mechanism will have minor influences. We will add a few sentences explaining this more explicitly in Line 397 after the Fig. 6 discussion. For example, we would add: "Note that both conditions may be present within a coarse-resolution pixel

because the pixel VOD spatially averages behavior over the landscape. Therefore, while plant resistance limitations may dominate landscapes that show 1-3-day VOD increases, slow infiltration responses may still be spatially prevalent, but to a lesser degree."

Reviewer 1: Minor comments Ln 160: Does this refer to the empirical probability mass function? Probably more clear/accurate to referr to as the histogram and described as a zero-inflated Poisson.

Authors: That is the correct and more specific distinction. We will revise line 160 to be: "The tp probability mass function within a given pixel typically has a mixed distribution with many zeros, resembling a zero-inflated poisson distribution."

Reviewer 1: Ln 176-184: This section could use greater clarity. Make clear that non-parametric ANOVAs are to compare the covariates of $\Delta$LAI, antecedent soil moisture, antecedent VOD, etc. between the three groups of VOD pulse responses.

Authors: We will add language to lines 180-184 for example in line 180: "The groups of three different tp lengths are then compared for each respective metric of ïĄĎLAI, antecedent surface soil moisture, soil moisture pulse magnitude, and antecedent VOD." Here, we make it clearer that the timescale groups are compared for each of the respective metrics.

Reviewer 1: Ln 231: ". . . aboveground biomass requires water uptake . . ."

Authors: This change will be made accordingly in line 231.

Reviewer 1: Ln 238: "co-occur"

Authors: This change will be made accordingly in line 238.

Reviewer 1: Ln 238-241: I suggest "Seasonal detrending of VOD isolates plant re-hydration response to moisture pulses, which show multi-day increases and eventual decreases following moisture inputs." Current phrasing is difficult to understand.

Authors: Based on this nice wording suggestion, we propose to change these lines to "Our seasonal detrending of VOD isolates these pulsed plant growth responses from seasonal growth cycles. These isolated sub-weekly VOD responses closely link to the timing of moisture pulses suggesting a cause-effect of rain pulse followed by plant water content response."

Reviewer 1: Section 3.3: Is this section also limited to Africa? Make clear at the outset, rather than in Ln 278-280. Other- wise, "growth-influenced VOD" is not supported.

Authors: These results are shown for Africa only. However, Fig. 5 is almost identical if shown globally. Therefore, we will change the plot to global, but make the note in line 289 that these results can still be directly compared one-to-one with Fig. 4 since the results are nearly identical if shown only for Africa.

Reviewer 1: Ln 299-306: This uncertainty analysis should be at least mentioned in the Methods section and lead with the purpose of this approach before referring the reader to the supplement.

Authors: We will add a new section 2.4 at line 190 that describes the uncertainty analysis. It will primarily consist of lines 299-306 with more explicit details of how the analysis was conducted. The first sentence will provide more explicit motivation.

Reviewer 1: Ln 326: Plant hydraulic capacitance, or something else?

Authors: Plant hydraulic capacitance is correct. We will add that.

Reviewer 1: Ln 332: ". . . pulse, which may indicate sufficiently well-watered conditions. . ."

Authors: We will make the update accordingly.

Reviewer 1: Ln 336-337: This sentence may not be necessary. Is the emphasis on parallel soil moisture and VOD decreases following the pulse? What is "plant-storage water potential"?

Authors: We agree that the sentence in its current location can be confusing. This sentence will be moved up to the second last line in line 334 and will be revised to: "Thereafter, VOD loss can be due to simultaneous soil and plant drying where plant rehydration becomes progressively restricted with drying soil." We will remove the "plant-storage water potential" phrase which was meant to refer to the collective water potential of the plant at predawn, but we see how it could be ambiguous.

Reviewer 1: Ln 339: Redefine RC time constants again at the be- ginning of the Discussion

Authors: We will add a brief definition in parenthesis at the end of line 339 that says "(plant water uptake and storage timescales)."

Reviewer 1: Ln 380: Again, I suggest introducing the SPAC model and its purpose in the Methods section before referring the reader to the supplement.

Authors: We will add a brief new methods section (Section 2.5) introducing the SPAC modeling methods and motivating the simulations. Here, we will refer the reader to more details in the SI.

Reviewer 1: Ln 406-408: This seems in direct contradiction to the above citation of Kramer and Boyer 1995. There is also the component of fine root growth after soil rewetting. Is the SPAC model able to represent the soil-root conductance or potential growth of new roots?

Authors: The Kramer and Boyer 1995 citation is in reference to a different point about soil infiltration potentially being faster and therefore giving evidence to root resistances being the driving factor here. It is not directly about root resistances as is the point in lines 406-408. We will remove the point about macropores in lines 395-397 because it is more speculative. We will add a note about fine root growth after rewetting in line 408: "Finally, fine root growth can occur after rewetting which can contribute to decreasing root resistances (Eissenstat et al., 1999)." The SPAC model does not include specific components due to soil-root resistance or root growth. We avoid more complex parameterizations and argue that the root resistance is sufficient to capture these dynamics discussed in lines 398-408 collectively though we don't have further information to partition the exact physiological mechanisms at large scales.

Reviewer 1: Ln 409-414: Please clarify the thoughts presented here. Is it to related VOD to water po- tential as well as water content? This paragraph seems out of place in this section and is not well-prefaced by the introduction.

Author: We have decided that we will remove this paragraph because it makes a subtle point that may be misleading in its current location. We will instead include a sentence in line 110 of the methods: "VOD is expected to be sensitive to rehydration because of near-linear relationships between relative water content and plant water potential, especially for herbaceous species which are primarily investigated in this study (Jones, 2014; Jones and Higgs, 1979; Konings et al., 2019; Nobel and Jordan, 1983)." Here, VOD's connection with relative water content has been established in a previous sentence and we are drawing the link between VOD and water potential.

Reviewer 1: Ln 433: ". . .demonstrating spatially-extensive evidence for. . ."

Author: We will update the text accordingly.

Reviewer 1: Ln 440: Use commas in the list

Author: We will update the text accordingly in line 440 with . . ."patterns of rehydration, growth responses, and rain pulse dependencies . . ."

Reviewer 1: Ln 444: "show" is used twice. Perhaps "We demonstrate. . .ecosystems exhibit . . ."

Author: We will update the text accordingly.

---

## Referee Comment (RC2) · Anonymous Referee #2 · 11 Nov 2020

Review of Feldman et al. "Patterns of plant rehydration and growth following pulses of soil moisture availability"

Feldman et al. use SMAP soil moisture and VOD data to show that plant response times to moisture pulses (characterized by "time-to-peak" between the start of the soil moisture drydown and peak VOD") are differentiated between humid regions (with tp of around zero) and dryland regions (tp >=1 days and up to >3). Furthermore, the authors use a satellite LAI dataset to distinguish between plant rehydration versus plant growth mechanisms for explaining the dryland VOD increase. From this latter analysis, they demonstrate that at shorter timescales (tp 1-3) the VOD increase is dominated by plant rehydration, and at longer timescales (tp >3) the VOD increase is more dominated by plant growth that occurs when antecedent conditions are wetter and the pulses of a

higher magnitude.

This study represents an important advance in our ability to remotely sense relatively short timescale vegetation responses to rainfall pulses, and further adds broader scale evidence to support the pulse-driven growth in dryland regions. The study context, questions, methods and results are clearly described and analyzed. The authors neatly address many caveats of the work, including limits and uncertainties associated to temporal sampling interval of the satellite instrument. I think this is exciting work that offers promise for exploring plant responses to moisture pulses in more depth when satellite sensor temporal sampling interval and spatial resolution are increased.

Main comments

My only remaining questions are relatively minor and are related to the LAI data. I appreciate the need to use a geostationary product given you require a high temporal resolution - thus, the choice of the EUMETSAT LSA SAF LAI product. Satellite LAI data are known to agree well in terms of temporal dynamics but differ in terms of absolute magnitude (e.g. Garrigues et al., 2008; Fang et al., 2013). I am wondering how biases in magnitude would impact your results, if at all? Perhaps the rates of change in LAI would be consistent across products. Did the authors look at any other optical geostationary satellite data products that are related to leaf growth/vegetation activity (NDVI, FAPAR, LAI)? I know there are not as many geostationary optical satellite products related to vegetation activity, and I confess I am not as familiar with these products, but I think there are some. For example, the GeoNEX products (Wang et al., 2020) and related to this I think is the NDVI from the AHI sensor (Miura et al., 2019). NDVI can be calculated from GOES-16/17 (https://www.ospo.noaa.gov/Products/land/vegetation.html).

A different point related to the LAI data: How noisy are the daily fluctuations? Are the changes in LAI you see after moisture pulses clearly detectable from the noise? Are the changes you see in LAI within the product uncertainty?

As the authors nicely discuss, these results are in line with many field-based studies. Therefore, I do not expect different LAI data, or any other optical satellite data related to leaf growth, to have a strong impact on the key findings here. I would just be curious as to how much of a difference the LAI (or NDVI etc) dataset makes and would be interested to see a brief discussion on any caveats related to LAI data noise and algorithm uncertainty.

I initially had a question has to why the LAI be decreasing across shorter tp timescales, and does that mean the positive changes in VOD actually reflect an even higher increase in plants' relative water content? This seems to be happening mostly in the Sahel. I then read in the discussion that this is because these events are mostly detected during periods when shrubs are shedding leaves, which makes sense given the shorter VOD increases are happening in drier periods. Have I understood correctly, or could there be any other reason? For longer duration tp > 3 the Sahel also has some decreases in LAI, with weaker increases than other regions in Africa. Is there any other reason to think the LAI in the Sahel is either less reliable or more influenced by other factors that are confounding these results?

Line 433-435: I am not sure this analysis fully supports this conclusion: "demonstrating evidence for the pulse-reserve hypothesis and suggesting sub-weekly (rain pulse) rather than seasonal phenological controls on growth (Noy-Meir, 1973)". As the authors have demonstrated, plant growth with longer tp periods are associated with wetter preceding conditions and stronger pulses. This could be in seasons that are already favorable for growth (as the authors state in lines 368-369), suggesting seasonal phenological controls (which may include temperature constraints) are still important. The pulses just result in that extra bit of growth. Given the studies that show inter-annual variability in net $CO_2$ uptake is strongly linked to days with peak gross $CO_2$ uptake (Zscheischler et al., 2016), I am wondering whether increases in leaf growth during the longer tp periods translate to increases in carbon uptake. Perhaps SIF data would be useful in this regard. However, this is probably beyond the scope of this study.

Minor comments

Fig. 4C is not referenced in the text. Fig. S5: describe sub-figure C in caption.

References

Fang, H., Jiang, C., Li, W., Wei, S., Baret, F., Chen, J. M., ... & Pinty, B. (2013). Characterization and intercomparison of global moderate resolution leaf area index (LAI) products: Analysis of climatologies and theoretical uncertainties. Journal of Geophysical Research: Biogeosciences, 118(2), 529-548.

Garrigues, S., Lacaze, R., Baret, F. J. T. M., Morisette, J. T., Weiss, M., Nickeson, J. E., ... & Knyazikhin, Y. (2008). Validation and intercomparison of global Leaf Area Index products derived from remote sensing data. Journal of Geophysical Research: Biogeosciences, 113(G2).

Miura, T., Nagai, S., Takeuchi, M., Ichii, K., & Yoshioka, H. (2019). Improved characterisation of vegetation and land surface seasonal dynamics in central Japan with Himawari-8 hypertemporal data. Scientific reports, 9(1), 1-12.

Wang, W., Li, S., Hashimoto, H., Takenaka, H., Higuchi, A., Kalluri, S., & Nemani, R. (2020). An Introduction to the Geostationary-NASA Earth Exchange (GeoNEX) Products: 1. Top-of-Atmosphere Reflectance and Brightness Temperature. Remote Sensing, 12(8), 1267.

Zscheischler, J., Fatichi, S., Wolf, S., Blanken, P. D., Bohrer, G., Clark, K., ... & Seneviratne, S. I. (2016). Short‐term favorable weather conditions are an important control of interannual variability in carbon and water fluxes. Journal of Geophysical Research: Biogeosciences, 121(8), 2186-2198.

---

## Author Comment (AC2) · 29 Nov 2020

Reviewer 2: Feldman et al. use SMAP soil moisture and VOD data to show that plant response times to moisture pulses (characterized by "time-to-peak" between the start of the soil moisture drydown and peak VOD") are differentiated between humid regions (with tp of around zero) and dryland regions (tp >=1 days and up to >3). Furthermore, the authors use a satellite LAI dataset to distinguish between plant rehydration versus plant growth mechanisms for explaining the dryland VOD increase. From this latter analysis, they demonstrate that at shorter timescales (tp 1-3) the VOD increase is dominated by plant rehydration, and at longer timescales (tp >3) the VOD increase is more dominated by plant growth that occurs when antecedent conditions are wetter and the pulses of a higher magnitude.

[Figure]

This study represents an important advance in our ability to remotely sense relatively short timescale vegetation responses to rainfall pulses, and further adds broader scale evidence to support the pulse-driven growth in dryland regions. The study context, questions, methods and results are clearly described and analyzed. The authors neatly address many caveats of the work, including limits and uncertainties associated to temporal sampling interval of the satellite instrument. I think this is exciting work that offers promise for exploring plant responses to moisture pulses in more depth when satellite sensor temporal sampling interval and spatial resolution are increased.

Authors: Thank you for your constructive comments. Please see our replies to your comments below.

Reviewer 2: Main comments My only remaining questions are relatively minor and are related to the LAI data. I appreciate the need to use a geostationary product given you require a high tem- poral resolution - thus, the choice of the EUMETSAT LSA SAF LAI product. Satel- lite LAI data are known to agree well in terms of temporal dynamics but differ in terms of absolute magnitude (e.g. Garrigues et al., 2008; Fang et al., 2013). I am wondering how biases in magnitude would impact your results, if at all? Perhaps the rates of change in LAI would be consistent across products.

Authors: The analysis makes use only of the temporal dynamics rather than the abso- lute magnitude. Figure 4 is evaluating the rates of change (time derivative). Therefore, subtracting out the mean value would produce the same results and similarly using the related fraction of vegetation cover product produces the same qualitative results as that in Figure 4. Furthermore, we are less concerned about the magnitude of LAI changes and more concerned about whether LAI is increasing or decreasing and how the sign of change influences the time to peak ($t_p$). We will add a new figure (proposed figure 4) as shown below in Fig. R1 to make this point clearer and to more clearly introduce original Fig. 4.

Reviewer 2: Did the authors look at any other optical geostationary satellite data products that are related to leaf growth/vegetation activity (NDVI, FAPAR, LAI)? I know there are not as many geostationary optical satellite products related to vegetation activity, and I confess I am not as familiar with these products, but I think there are some. For example, the GeoNEX products (Wang et al., 2020) and related to this I think is the NDVI from the AHI sensor (Miura et al., 2019). NDVI can be calculated from GOES-16/17 (https://www.ospo.noaa.gov/Products/land/ vegetation.html).

Authors: To the authors knowledge, there are no other geostationary satellites that would provide a one-to-one comparison with the MSG-SEVIRI satellite we are using in Africa. The MSG series is included in the GeoNEX products mentioned here.

The AHI Sensor covers primarily Japan and other Pacific Islands which will not provide a one-to-one comparison with Fig. 4 and also occurs in a region where the microwave data are less available due to radio frequency interference (quality flags due to interference from telecommunication).

The GOES-16/17 provides observations over primarily the North American region which does not provide a one-to-one comparison with results found in Africa and a comparison between Africa and North America behavior would not provide clear implications about LAI uncertainty. It may instead provide insights into the less-spatially extensive Southwest US dryland regions relative to African drylands and potential differences in responses of the biomes therein.

Note that we repeated Fig. 4 using SEVIRI FAPAR which provides similar qualitative results. See Fig. R2 below. While FAPAR is retrieved from the same satellite, it uses different frequencies of measurements and a different algorithmic approach than for LAI such that is provides at least a partially independent vegetation index from that of LAI.

Reviewer 2: A different point related to the LAI data: How noisy are the daily fluctuations? Are the changes in LAI you see after moisture pulses clearly detectable from the noise? Are the changes you see in LAI within the product uncertainty?

Authors: In the SEVIRI LAI data, a filter is applied to mitigate cloud cover contamination: previous days before the measurement are nonlinearly averaged where the most recent days contribute more to the displayed value. Daily fluctuations are therefore inherently smoothed. The post-processing smoothing technique thus obscures this error/uncertainty estimation. The available uncertainty metrics provided from LandSAF are related to the magnitude of the LAI measurements and bias compared to ground measurements rather than a more desirable error standard deviation estimate for this application here, which would quantify how much of a daily LAI change is due to noise. We are not aware of such standard deviation LAI estimates and could not find any available in the literature.

Ultimately, we are most concerned with whether SEVIRI is confidently estimating the LAI seasonal cycle. Even if there is an uncertain dLAI/dt value on a given day, we are more interested qualitatively if this value is positive, negative, or near zero during a given soil moisture drydown. Therefore, as long as the seasonal cycle of LAI is well resolved with these daily measurements, we can be confident that a dLAI is capturing a growth or senescence stage. The proposed new Fig. 4 reflects this idea (see Fig. R1 above). We are confident in our dLAI/dt estimates about the season changes in LAI for several reasons:

1) SEVIRI LAI samples effectively every 3-5-days compared to low Earth orbit satellites like MODIS which effectively sample every 15-20 days (Fensholt et al. 2006). This is due to cloud cover contamination which obscures the SEVIRI 15 minutes actual sampling and the MODIS 1-2 day actual sampling. Based on this consideration, SEVIRI is better able to resolve the seasonal cycle, especially during the wet season (of most interest for detecting the tp>3) compared to noisier measurements from low Earth orbit (MODIS) (Gessner et al. 2013). See Fig. R3 below. 2) We are estimating average dLAI/dt over ~10 day stages (during soil moisture drydowns), which become insensitive to whether or not a given day change of LAI is detectable above a noise level. We are thus confident in determining whether a 10 day trend in LAI is positive or negative,

especially given that SEVIRI LAI is able to resolve the seasonal cycle. 3) Uncertainties are lowest in Africa due to its lower view angle and especially in regions with more herbaceous vegetation (Garcia-Herrero et al. 2013). Therefore, the regions we are evaluating (see Fig. 4) generally have the lowest errors compared to other regions measured by SEVIRI.

In the reply to the next comment, we propose changes to specific sections.

Fensholt, R., Sandholt, I., Stisen, S., Tucker, C., 2006. Analysing NDVI for the African continent using the geostationary meteosat second generation SEVIRI sensor. Remote Sens. Environ. 101, 212–229. https://doi.org/10.1016/j.rse.2005.11.013 García-Haro, F.J., Camacho, F., Meliá, J., 2013. The EUMETSAT Satellite Application Facility on Land Surface Analysis Product User Manual Vegetation Parameters (VEGA) 401, 1–46. Gessner, U., Niklaus, M., Kuenzer, C., Dech, S., 2013. Intercomparison of leaf area index products for a gradient of sub-humid to arid environments in west africa. Remote Sens. 5, 1235–1257. https://doi.org/10.3390/rs5031235

Reviewer 2: As the authors nicely discuss, these results are in line with many field-based studies. Therefore, I do not expect different LAI data, or any other optical satellite data related to leaf growth, to have a strong impact on the key findings here. I would just be curious as to how much of a difference the LAI (or NDVI etc) dataset makes and would be interested to see a brief discussion on any caveats related to LAI data noise and algorithm uncertainty.

Authors: As stated in our replies to the previous comments, there is not another feasible LAI dataset to directly compare with the SEVIRI LAI data in this study in Africa. Another geostationary product could be used in another region and it would be unclear whether differences would be due to vegetation/climate or LAI dataset differences. Low Earth orbit satellites like MODIS are not feasible for this application since they are unable to resolve the seasonal cycle, especially during the wet season (see our reply to the previous comment).

We will revise and extend the methods paragraph in lines 120-130 to add a more comprehensive discussion of the LAI data and its uncertainties that makes the following points: That we are confident in LAI detecting increases over 1-2 weeks because: (1) SEVIRI LAI resolves the seasonal cycle of growth and senescence stages well due to rapid sampling and filtering techniques which can resolve the seasonal cycle even during the wet season. See an example in Fig. R3. (2) Because of (1), we are confident we can detect the stage of the LAI seasonal trend over a 1-2-week period which less insensitive to uncertainties in daily LAI rates of change (3) LAI retrievals in herbaceous biomes of Africa, evaluated here, have the lowest uncertainty (4) Use of LAI data is primarily for qualitative purposes (increase/decrease) which makes our analysis less sensitive to noise considerations at daily scales and to biases in absolute LAI magnitudes.

We will add a statement such as: "Ultimately, we are more interested in qualitatively increasing trends in LAI rather than the magnitudes of LAI rates of change which are less certain." in line 253 to denote that we are primarily focused on qualitative increases rather than LAI magnitudes. We will also add a statement about why we are confident in the trends in the discussion in Line 350 that reflect the points above as well. Finally, we will add a new figure as new Figure 4 (Fig. R1), that shows how we are using the LAI data primarily as a binary metric as to whether tp is estimated during a growth period.

Reviewer 2: I initially had a question has to why the LAI be decreasing across shorter tp timescales, and does that mean the positive changes in VOD actually reflect an even higher in- crease in plants' relative water content? This seems to be happening mostly in the Sahel. I then read in the discussion that this is because these events are mostly detected during periods when shrubs are shedding leaves, which makes sense given the shorter VOD increases are happening in drier periods. Have I understood correctly, or could there be any other reason?

Authors: Your interpretation is consistent with ours that a decrease in LAI/biomass

would suggest that the positive increase in relative water content reflected in VOD is likely being subdued by the biomass decrease. We will extend our statement in line 375 to read: "They can occur even when biomass is decreasing (Fig. 4; such as leaf off), where the relative water content increases are likely larger than what the raw VOD increasing signal suggests."

Reviewer 2: For longer duration tp > 3 the Sahel also has some decreases in LAI, with weaker increases than other regions in Africa. Is there any other reason to think the LAI in the Sahel is either less reliable or more influenced by other factors that are confounding these results?

Authors: Based on various in-situ assessments and validation reports, there is no evidence provided that the LAI uncertainty in the Sahel would be less certain than LAI measurements in similar climates of Southern Africa. Furthermore, soil contamination mitigation techniques (with a Gaussian mixture model) are implemented within the LAI algorithm to reduce sources of error from bare soil common in less vegetated regions such as the Sahel.

We assessed the pixels in the Sahel where LAI decreases during tp>3 on average. These appear to be related to LAI seasonal cycles where, in this specific region, dLAI/dt differentially has a greater magnitude slope (absolute value) during the senescence stage than the dLAI/dt during the growth stage. There are large negative dLAI/dt during tp>3 detected periods during the senescence stage that then bias the overall mean dLAI/dt estimate. This does not occur in nearby pixels in the Sahel or in similar climates of Southern Africa where LAI increases during tp>3 on average. See Fig. R4 below for an example of this scenario where the blue line shows how the dLAI/dt decreases are differentially larger in magnitude than the increases in these regions in question in the Sahel. This does not occur in other regions (such as for the red and green lines). We ultimately could not determine a method to objectively remove cases of these LAI decreases.

Ultimately, we are attempting to quantify the overall trends in growth and avoid interpreting specific pixels, acknowledging that there that there will be cases of noisy tp estimates with the VOD time series. For example, there are may be cases where a VOD increase was truly tp = 2 days but a single noisy observation created a tp = 5 day estimate. In taking the average over the pixel, we intend to detect the mean responses amongst uncertain estimates. See our section 3.4 and specifically lines 318-319. Therefore, we anticipate biases in even full pixel estimates of tp which may result in tp>3 falsely related to decreases in LAI. Based on these considerations, we avoid interpreting results in specific pixels. We will make this point clearer in line 231.

Reviewer 2: Line 433-435: I am not sure this analysis fully supports this conclusion: "demonstrating evidence for the pulse-reserve hypothesis and suggesting sub-weekly (rain pulse) rather than seasonal phenological controls on growth (Noy-Meir, 1973)". As the authors have demonstrated, plant growth with longer tp periods are associated with wetter preceding conditions and stronger pulses. This could be in seasons that are already favorable for growth (as the authors state in lines 368-369), suggesting seasonal phenological controls (which may include temperature constraints) are still important. The pulses just result in that extra bit of growth.

Authors: This is an excellent point. As the reviewer points out, we do acknowledge that there is a phenological component to the growth in lines 368-369. Therefore, the statement about phenological controls in line 433 is a misstatement of the results. We propose to change the sentence in line 433 in the conclusions to be more consistent with our discussion statements that are similar to the point made by the reviewer: "Therefore, dryland vegetation intermittently upregulates and grows after individual rainfall events, demonstrating spatially-extensive evidence for the pulse-reserve hypothesis. Specifically, we show that there is a component of growth linked directly to individual rainfall events in addition to any continuous seasonal growth (Noy-Meir, 1973)."

Reviewer 2: Given the studies that show inter-annual variability in net CO2 uptake is strongly linked to days with peak gross CO2 uptake (Zscheischler et al., 2016), I am

wondering whether increases in leaf growth during the longer tp periods translate to increases in carbon uptake. Perhaps SIF data would be useful in this regard. However, this is probably beyond the scope of this study.

Authors: While evaluating carbon fluxes/photosynthesis is beyond the scope of this study, we have another study under review and likely to be published in time to reference in this paper. In that paper, we show that carbon fluxes at flux towers show similar signatures to VOD responses here, specifically that the greatest and longer duration increases in NEP occur after larger rainfall events on wetter surfaces.

Reviewer 2: Minor comments Fig. 4C is not referenced in the text.

Authors: Good catch. We will mention 4C in the parentheses in addition to 4A and 4D in line 228. We will also denote 4C in the parentheses in line 375.

Reviewer 2: Fig. S5: describe sub-figure C in caption.

Authors: Good catch. In Fig. S5 caption, we will add: "(C) Incorporating random noise into the algorithm appears to increase false detection of non-zero tp the most. Ultimately, all effects together still result in frequent correct detection of true tp of zero."
* * *
[Figure]

[Figure]

**Fig. 1.** Fig. R1. Proposed new Fig. 4 in the manuscript. Growth increases the timescale of plant water content in African regions with median tp $\geq$1 day. Medians of the two bins are significantly different (p<0

[Figure]

**Fig. 2.** Fig. R2. Figure 4 repeated with FAPAR which is derived from a separate algorithmic approach in LandSAF.

[Figure]

**Fig. 3.** Fig. R3. Example Sahel Grasslands pixel with seasonal cycle of SEVIRI and MODIS LAI. MODIS is less able to resolve the seasonal cycle during the wet season due to less frequent sampling than SEVIRI.

[Figure]

**Fig. 4.** Fig. R4. LAI annual time series at example pixels where dLAI/dt was positive and negative.

---

## Author Response (AR1)

**Responses to Co-Editor-in-Chief**

| Reviewer Comments | Response |
|---|---|
| **Comment 1:**
**Dear Authors,**

**Thank you for your careful consideration of the reviewers comments. The associate editor is also supportive of your work, but I would encourage you to take their comments on board when preparing your revision.**

**Best wishes,**
**Trevor** | Thank you for your comment. We have carefully considered the associate editor's comments below as well. |

**Responses to Associate Editor**

| Reviewer Comments | Response |
|---|---|
| **Comments to the Author:**
**Dear Authors,**

**I have received two positive reviews of your work and I am recommending that your work should be published subject to minor revisions. I am happy with the proposed changes in response to the reviewer's comments.**

**I was particularly interested in your work because I had attempted to do something similar, but using changes in surface temperature (https://journals.ametsoc.org/jhm/article /14/5/1605/206524/Quantifying-Land-Surface-Temperature-Variability). When you use the SPAC (fig 6) to diagnose differences I have to say I tended to agree with R1. I myself was wondering about the spatial scales of the data and multiple, fine-scale contributions to your results. I have no problem with the SPAC analysis, so do not feel you need a change anything, but you might consider reflecting that data coarseness/aggregation may smear dry-down interpretation too. A sentence on data resolution may warrant a sentence if you feel it is appropriate.**

**In your response to R2, you make clear that the LAI magnitude itself isn't that important, but rather the direction of change, might it be worth emphasising this point in the manuscript too for** | Thank you for your constructive comments on our study.

In response to your thoughts on reviewer 1 comments, see our reply to reviewer 1 comment 4. We agree that the spatial aggregation of heterogeneous sub-pixel behavior is worth mentioning. We have added a short discussion in lines 446-450 that discusses how responses at finer scales can be distorted when aggregating spatially and acknowledge that both types of behavior discussed in Fig. 7 are simultaneously possible within the same pixel. We also add acknowledgement of the aggregated behavior directly in discussing Fig. 3 in line 240.

In response to your thoughts on reviewer 2 comments, see our replies to reviewer 2 comments 3, 4, and 5. In summary, we have added that we are more interested in the LAI rate of change rather than LAI magnitude as well as added the robustness to FAPAR in an extended statement on the LAI dataset in the methods in lines 134-145. In addition, we have added Fig. R1 below in the supplemental materials as Fig. S1 consistent with your suggestion.

Thank you for your reference to De Kauwe et al. (2013) which is also quite helpful in reference to our ongoing work in using SEVIRI to study daily land-atmosphere coupling in Africa in a similar context as the study here. |

| other readers? Similarly, you note that your results were broadly robust to the use of fapar instead, why not again note this for the reader. Perhaps you might also include a version of fig 4 in the supplement (optional)?

Best wishes,

Martin De Kauwe | |

**Important:** Please note that line numbers in reviewer comments in the left column are in reference to the **initial submission**. The line numbers in author response in the right column are in reference to the revised manuscript submission.

| Reviewer Comments | Response |
|---|---|
| **Comment 1:**
**This study assesses the temporal pattern of soil moisture and plant water content derived from SMAP to understand plant responses to precipitation pulses, which have shown significant lags in field studies. The comprehensive spatial scale of this analysis is impressive and timely. Importantly, the authors use a daily LAI product for Africa to account for the combined growth and rehydration signal inherent in VOD interpretation. I have some concerns about the correct use of Kruskal-Wallis tests and found the comparison between spatially average temporal VOD trends and results of a 1D SPAC model somewhat spurious.** | Thank you for your constructive comments on our study. |
| **Comment 2:**
**Major comments**
**Ln 282 - 287: This section is throwing me off because I cannot easily distinguish between rapid responses and short VOD increases; they sound the same to me. Adding the actual tp lengths would help.** | We have added in parentheses in line 330 that rapid VOD responses have tp = 0 days and short VOD increases have tp of 1-3 days. |
| **Comment 3:**
**More importantly, the boxplots of Fig. 5 do not convey distinctions based on the KW tests. Is there some kind of non-parametric post-hoc test, equivalent to Tukey HSD, which can be used to put labels on the boxplots? As far as I am aware, the KW tests only indicate that at least one group is significantly different but does not indicate which ones. This may require pairwise Mann-Whitney tests.** | It is correct that the KW test does not identify which groups exhibit significant differences. We computed all possible pairwise Mann-Whitney U tests between all variables and found statistically significant ($p<0.05$) differences for all tests. We wish to avoid adding labels to the boxplot. Therefore, we have noted that pairwise Mann Whitney U tests show significant differences between all groups in lines 342-343 in the Figs. 5 and 6 captions. |
| **Comment 4:**
**I don't think the spatially averaged time series shown in Fig. 3A and B are comparable to a 1D SPAC model result in Fig. 6A and B. Rather, both kinds of modeled drydown patterns might be found in the actual data if not aggregated spatially. The relatively occurrence of imme-** | We acknowledge the differences in spatial scale of the results. We may not have been explicit about our point: that the spatially dominant mechanism will influence the VOD signal the most, while other mechanisms will have more minor influences. Note that we are only discussing cases where satellite plant water content responses are 1-3 days, not across all |

| | |
|---|---|
| **diate versus lagged rehydration may additionally depend on antecedent soil moisture conditions.** | timescales as in Fig. 3. We have updated the text in lines 446-450 to make this point clearer. Here, we have added a few sentences explaining the effect of aggregating heterogeneous responses that depend on antecedent moisture heterogeneity at scales finer than the grid scale after the Fig. 7 discussion. |
| **Comment 5:**
**Minor comments Ln 160: Does this refer to the empirical probability mass function? Probably more clear/accurate to referr to as the histogram and described as a zero-inflated Poisson.** | That is the correct and more specific distinction. We have revised lines 177-178 to be: "The $t_p$ probability mass function within a given pixel typically has a mixed distribution with many zeros, resembling a zero-inflated poisson distribution." |
| **Comment 6:**
**Ln 176-184: This section could use greater clarity. Make clear that non-parametric ANOVAs are to compare the covariates of ΔLAI, antecedent soil moisture, antecedent VOD, etc. between the three groups of VOD pulse responses.** | We have added language to lines 197-201 to make it clearer that the timescale groups are compared for each of the respective metrics. |
| **Comment 7:**
**Ln 231: ". . . aboveground biomass requires water uptake . . ."** | This change has been made in line 273. |
| **Comment 8:**
**Ln 238: "co-occur"** | This change has been made in line 282. |
| **Comment 9:**
**Ln 238-241: I suggest "Seasonal detrending of VOD isolates plant rehydration response to moisture pulses, which show multi-day increases and eventual decreases following moisture inputs." Current phrasing is difficult to understand.** | Based on this nice wording suggestion, we have revised this statement in lines 283-285 to be: "Our seasonal detrending of VOD isolates these pulsed plant growth responses from seasonal growth cycles. These isolated sub-weekly VOD responses closely link to the timing of moisture pulses suggesting a cause-effect of rain pulse followed by plant water content response." |
| **Comment 10:**
**Section 3.3: Is this section also limited to Africa? Make clear at the outset, rather than in Ln 278-280. Other- wise, "growth-influenced VOD" is not supported.** | These results were originally shown for Africa only. However, Fig. 6 (previously Fig. 5) is almost identical if shown globally. Therefore, we have changed the plot in Fig. 6 to global, but make the note in lines 327-329 that these results can still be directly compared one-to-one with Fig. 5 (which is only for Africa) since the results are nearly identical if shown only for Africa. |
| **Comment 11:**
**Ln 299-306: This uncertainty analysis should be at least mentioned in the Methods section and lead with the purpose of this approach before referring the reader to the supplement.** | We have added a new section 2.4 at line 210 that describes the uncertainty analysis with more explicit details of how the analysis was conducted. The first sentence in line 210 provides more explicit motivation. |

| | |
|---|---|
| **Comment 12:**
**Ln 326: Plant hydraulic capacitance, or something else?** | Plant hydraulic capacitance is correct. We have made this revision in line 371. |
| **Comment 13:**
**Ln 332: ". . . pulse, which may indicate sufficiently well-watered conditions. . ."** | We have updated the text accordingly in line 379. |
| **Comment 14:**
**Ln 336-337: This sentence may not be necessary. Is the emphasis on parallel soil moisture and VOD decreases following the pulse? What is "plant-storage water potential"?** | We agree that the sentence in its current location can be confusing. This sentence has been reduced and revised for clarity and has been moved up to line 375. We have removed discussion of "plant-storage water potential" here, but revise its definition in the Fig. 7 caption to be predawn water potential of the plant water stores in line 467. |
| **Comment 15:**
**Ln 339: Redefine RC time constants again at the be- ginning of the Discussion** | We have added a brief definition in parenthesis at the end of this sentence in line 384 that says "(plant water uptake and storage timescales)." |
| **Comment 16:**
**Ln 380: Again, I suggest introducing the SPAC model and its purpose in the Methods section before referring the reader to the supplement.** | We have added a brief new methods section (Section 2.5) introducing the SPAC modeling methods and motivating the simulations starting on line 223. We have referred the reader to more details in the SI in line 230. |
| **Comment 17:**
**Ln 406-408: This seems in direct contradiction to the above citation of Kramer and Boyer 1995. There is also the component of fine root growth after soil rewetting. Is the SPAC model able to represent the soil-root conductance or potential growth of new roots?** | The Kramer and Boyer 1995 reference is in reference to a different point about soil infiltration potentially being faster and therefore giving evidence for root resistances being the driving factor here. It is not directly about root resistances as is the point in original manuscript lines 406-408. We have removed the point about macropores in original manuscript lines 395-397 because it is more speculative.

We have added a note about fine root growth after rewetting in line 461.

The SPAC model does not include specific components due to soil-root resistance or root growth. We avoid more complex parameterizations and argue that the root resistance is sufficient to capture these dynamics discussed in lines 451-461 collectively though we don't have further evidence to partition the exact physiological mechanisms at large scales. |
| **Comment 18:**
**Ln 409-414: Please clarify the thoughts presented here. Is it to related VOD to water po- tential as well as water content? This paragraph seems out** | We have decided to remove this paragraph because it makes a subtle point that may be misleading in its current location. We have instead included a sentence (summarizing the points in the deleted paragraph) in line 110 of the methods. Here, VOD's connection |

| | |
|---|---|
| **of place in this section and is not well-prefaced by the introduction.** | with relative water content has been established in a previous sentence and we are drawing the link between VOD and water potential. |
| **Comment 19:**
**Ln 433: ". . .demonstrating spatially-extensive evidence for. . ."** | We have updated the text in line 482. |
| **Comment 20:**
**Ln 440: Use commas in the list** | We have updated the text accordingly in line 489. |
| **Comment 21:**
**Ln 444: "show" is used twice. Perhaps "We demonstrate. . .ecosystems exhibit . . ."** | We have updated the text accordingly in line 494. |

**Responses to Reviewer #2**

**Important:** Please note that line numbers in reviewer comments in the left column are in reference to the **initial submission**. The line numbers in author response in the right column are in reference to the revised manuscript submission.

| Reviewer Comments | Response |
|---|---|
| **Comment 1:** | |
| **Feldman et al. use SMAP soil moisture and VOD data to show that plant response times to moisture pulses (characterized by "time-to-peak" between the start of the soil moisture drydown and peak VOD") are differentiated between humid regions (with tp of around zero) and dryland regions (tp >=1 days and up to >3). Furthermore, the authors use a satellite LAI dataset to distinguish between plant rehydration versus plant growth mechanisms for explaining the dryland VOD increase. From this latter analysis, they demonstrate that at shorter timescales (tp 1-3) the VOD increase is dominated by plant rehydration, and at longer timescales (tp >3) the VOD increase is more dominated by plant growth that occurs when antecedent conditions are wetter and the pulses of a higher magnitude.** | Thank you for your constructive comments. Please see our replies to your comments below. |
| **This study represents an important advance in our ability to remotely sense relatively short timescale vegetation responses to rainfall pulses, and further adds broader scale evidence to support the pulse-driven growth in dryland regions. The study context, questions, methods and results are clearly described and analyzed. The authors neatly address many caveats of the work, including limits and uncertainties associated to temporal sampling interval of the satellite instrument. I think this is exciting work that offers promise for exploring plant responses to moisture pulses in more depth when satellite sensor temporal sampling** | |

| | |
|---|---|
| **interval and spatial resolution are increased.** | |
| **Comment 2:**
**Main comments**
**My only remaining questions are relatively minor and are related to the LAI data. I appreciate the need to use a geostationary product given you require a high tem- poral resolution - thus, the choice of the EUMETSAT LSA SAF LAI product. Satel- lite LAI data are known to agree well in terms of temporal dynamics but differ in terms of absolute magnitude (e.g. Garrigues et al., 2008; Fang et al., 2013). I am wondering how biases in magnitude would impact your results, if at all? Perhaps the rates of change in LAI would be consistent across products.** | The analysis makes use only of the temporal dynamics rather than the absolute magnitude. Figure 4 (in the original manuscript) is evaluating the time rates of change. Therefore, subtracting out the mean value would produce the same results and similarly using the related fraction of absorbed photosynthetically active radiation (FAPAR) product produces the same qualitative results as that in Figure 4. Furthermore, we are less concerned about the magnitude of LAI changes and more concerned about whether LAI is increasing or decreasing and how the sign of change influences the time to peak (tp). We have added a new figure (Fig. R1 below which is now figure 4) to make this point clearer. See updated text in reference to new Fig. 4 in line 267.

[Figure]

Fig. R1. (now Fig. 4 in manuscript). Relationship of plant water content increase timescales with biomass changes in African regions with median tp ≥1 day. Growth increases the timescale of plant water content. Mann-Whitney U tests indicate that the medians of the two bins are significantly different (p<0.05). |
| **Comment 3:**
**Did the authors look at any other optical geostationary satellite data products that are related to leaf growth/vegetation activity (NDVI, FAPAR, LAI)? I know there are not as many geostationary optical satellite products related to vegetation activity, and I confess I am not as familiar with** | To the authors knowledge, there are no other geostationary satellites that would provide a one-to-one comparison with the MSG-SEVIRI satellite we are using in Africa. The MSG series is included in the GeoNEX products mentioned here.

The AHI Sensor covers primarily Japan and other Pacific Islands which will not provide a one-to-one comparison with Figs. 4 and 5 and also occurs in a region where the microwave |

| | |
|---|---|
| **these products, but I think there are some. For example, the GeoNEX products (Wang et al., 2020) and related to this I think is the NDVI from the AHI sensor (Miura et al., 2019). NDVI can be calculated from GOES-16/17 ([https://www.ospo.noaa.gov/Products/land/](https://www.ospo.noaa.gov/Products/land/) vegetation.html).** | data are less available due to radio frequency interference (quality flags due to interference from telecommunication).

The GOES-16/17 provides observations over primarily the North American region which does not provide a one-to-one comparison with results found in Africa and a comparison between Africa and North America behavior would not provide clear implications about LAI uncertainty. It may instead provide insights into the less-spatially extensive Southwest US dryland regions relative to African drylands and potential differences in responses of the biomes therein.

Note that we repeated Fig. 5 using SEVIRI FAPAR which provides similar qualitative results. See Fig. R2 below. While FAPAR is retrieved from the same satellite, it uses different frequencies of measurements and a different algorithmic approach than for LAI such that is provides at least a partially independent vegetation index from that of LAI.

We have included a more explicit statement that other available satellites that sample globally are too coarse for our applications here in lines 136-143. We have also added a statement about reimplementing the analysis on FAPAR as shown in Fig. R2 below in lines 143-145 and line 289 and included it as Fig. S1.

[Figure]

Fig. R2. Figure 5 (originally Figure 4) repeated with FAPAR which is derived from a separate algorithmic approach in LandSAF. Now Fig. S1. |
| **Comment 4:**
**A different point related to the LAI data: How noisy are the daily** | In the SEVIRI LAI data, a filter is applied to mitigate cloud cover contamination: previous days before the measurement are nonlinearly averaged where the most recent days contribute |

| | |
|---|---|
| **fluctuations? Are the changes in LAI you see after moisture pulses clearly detectable from the noise? Are the changes you see in LAI within the product uncertainty?** | more to the displayed value. Daily fluctuations are therefore inherently smoothed. The post-processing smoothing technique thus obscures this error/uncertainty estimation. The available uncertainty metrics provided from LandSAF are related to the magnitude of the LAI measurements and bias compared to ground measurements rather than a more desirable error standard deviation estimate for this application here, which would quantify how much of a daily LAI change is due to noise. We are not aware of such standard deviation LAI estimates and could not find any available in the literature.

Ultimately, we are most concerned about whether SEVIRI is confidently estimating the LAI seasonal cycle. Even if there is an uncertain dLAI/dt value on a given day, we are more interested qualitatively if this value is positive, negative, or near zero during a given soil moisture drydown. Therefore, as long as the seasonal cycle of LAI is well resolved with these daily measurements, we can be confident that a dLAI/dt value is capturing a growth or senescence stage. The new Fig. 4 reflects this idea (see Fig. R1 above). We are confident in our dLAI/dt estimates about the seasonal changes in LAI for several reasons:

1) SEVIRI LAI samples effectively every 3-5-days compared to low Earth orbit satellites like MODIS which effectively sample every 15-20 days (Fensholt et al. 2006). This is due to cloud cover contamination which obscures the SEVIRI 15 minutes actual sampling and the MODIS 1-2 day actual sampling. Based on this consideration, SEVIRI is better able to resolve the seasonal cycle, especially during the wet season (of most interest for detecting the tp>3) compared to noisier measurements from low Earth orbit (MODIS) (Gessner et al. 2013). See Fig. R3 below.
2) We are estimating average dLAI/dt over ~10 day stages (during soil moisture drydowns), which become insensitive to whether or not a given day change of LAI is detectable above a noise level. We are thus confident in determining whether a 10 day trend in LAI is positive or negative, especially given that SEVIRI LAI is able to resolve the seasonal cycle.
3) Uncertainties are lowest in Africa due to its lower view angle and especially in regions with more herbaceous vegetation (Garcia-Herrero et al. 2013). Therefore, the regions we are evaluating (see Fig. 4) generally have the lowest errors compared to other regions measured by SEVIRI.

These points are now all reflected in new lines 136-145 in the methods. |

[Figure]

Fig. R3. Example pixel with seasonal cycle of SEVIRI and MODIS LAI at a location in the Sahel Grasslands. MODIS is less able to resolve the seasonal cycle especially during the wet season due to less frequent sampling than SEVIRI.

Fensholt, R., Sandholt, I., Stisen, S., Tucker, C., 2006. Analysing NDVI for the African continent using the geostationary meteosat second generation SEVIRI sensor. Remote Sens. Environ. 101, 212–229. https://doi.org/10.1016/j.rse.2005.11.013

García-Haro, F.J., Camacho, F., Meliá, J., 2013. The EUMETSAT Satellite Application Facility on Land Surface Analysis Product User Manual Vegetation Parameters (VEGA) 401, 1–46.

Gessner, U., Niklaus, M., Kuenzer, C., Dech, S., 2013. Intercomparison of leaf area index products for a gradient of sub-humid to arid environments in west africa. Remote Sens. 5, 1235–1257. https://doi.org/10.3390/rs5031235

**Comment 5:**
**As the authors nicely discuss, these results are in line with many field-based studies. Therefore, I do not expect different LAI data, or any other optical satellite data related to leaf growth, to have a strong impact on the key findings here. I would just be curious as to how much of a difference the LAI (or NDVI etc) dataset makes and would be interested to see a brief discussion on any caveats related to LAI data noise and algorithm uncertainty.**

As stated in our replies to the previous comments, there is not another feasible LAI dataset to directly compare with the SEVIRI LAI data in this study in Africa. Another geostationary product could be used in another region and it would be unclear whether differences would be due to vegetation/climate or LAI dataset differences. Low Earth orbit satellites like MODIS are not feasible for this application since they are unable to resolve the seasonal cycle, especially during the wet season (see our reply to the previous comment).

We have revised and extended the methods paragraph in lines 136-145 to add a more comprehensive discussion of the LAI data and its uncertainties that makes the following points:

| | That we are confident in LAI detecting increases over 1-2 weeks because: |
|---|---|
| |     (1) SEVIRI LAI resolves the seasonal cycle of growth and senescence stages well due to rapid sampling and filtering techniques which can resolve the seasonal cycle even during the wet season. See an example in Fig. R3.
     (2) Because of (1), we are confident we can detect the stage of the LAI seasonal trend over a 1-2-week period which less insensitive to uncertainties in daily LAI rates of change
     (3) LAI retrievals in herbaceous biomes of Africa, evaluated here, have the lowest uncertainty
     (4) Use of LAI data is primarily for qualitative purposes (increase/decrease) which makes our analysis less sensitive to noise considerations at daily scales and to biases in absolute LAI magnitudes.

 We have additionally added a new statement in line 134 that we are primarily focused on qualitative increases rather than LAI magnitudes. We also added a statement about why we are confident in the trends in the discussion in lines 396-399 that reflect the points above as well. Finally, as mentioned in our reply to comment 2, we have added a new Figure 4 that shows how we are using the LAI data primarily as a binary metric as to whether tp is estimated during a growth period. |
| **Comment 6:**
 **I initially had a question has to why the LAI be decreasing across shorter tp timescales, and does that mean the positive changes in VOD actually reflect an even higher in- crease in plants' relative water content? This seems to be happening mostly in the Sahel. I then read in the discussion that this is because these events are mostly detected during periods when shrubs are shedding leaves, which makes sense given the shorter VOD increases are happening in drier periods. Have I understood correctly, or could there be any other reason?** | Your interpretation is consistent with ours that a decrease in LAI/biomass would suggest that the positive increase in relative water content reflected in VOD is likely being subdued by the biomass decrease. We have updated our statement in line 426 to reflect this concept. |
| **Comment 7:**
 **For longer duration tp > 3 the Sahel also has some decreases in LAI, with weaker increases than other regions in Africa. Is there any other reason to think the LAI in the Sahel is either less reliable or more influenced by other** | Based on various in-situ assessments and validation reports, there is no evidence provided that the LAI uncertainty in the Sahel would be less certain than LAI measurements in similar climates of Southern Africa. Furthermore, soil contamination mitigation techniques (with a Gaussian mixture model) are implemented within the LAI algorithm to reduce sources of error from bare soil common in less vegetated regions such as the Sahel. |

| **factors that are confounding these results?** | We assessed the pixels in the Sahel where LAI decreases during tp>3 on average. These appear to be related to LAI seasonal cycles where, in this specific region, dLAI/dt differentially has a greater magnitude slope (absolute value) during the senescence stage than the dLAI/dt during the growth stage. There are large negative dLAI/dt during tp>3 detected periods during the senescence stage that then bias the overall mean dLAI/dt estimate. This does not occur in nearby pixels in the Sahel or in similar climates of Southern Africa where LAI increases during tp>3 on average. See Fig. R4 below for an example of this scenario where the blue line shows how the dLAI/dt decreases are differentially larger in magnitude than the increases in these regions in question in the Sahel. This does not occur in other regions (such as for the red and green lines). We could not determine a method to objectively remove cases of these LAI decreases. |
| :--- | :--- |
| | Ultimately, we are attempting to quantify the overall trends in growth and avoid interpreting specific pixels, acknowledging that there that there will be cases of noisy tp estimates with the VOD time series. For example, there are may be cases where a VOD increase was truly tp = 2 days but a single noisy observation created a tp = 5 day estimate. In taking the average over the pixel, we intend to detect the mean responses amongst uncertain estimates. See our section 3.4 and specifically lines 363-366. Therefore, we anticipate biases in even full pixel estimates of tp which may result in tp>3 falsely related to decreases in LAI. Based on these considerations, we avoid interpreting results in specific pixels. |
| | We made these points clearer in new lines 273-276. |
| |
[Figure]
 |
| | Fig. R4. LAI annual time series at example pixels where dLAI/dt was positive and negative. |

| | |
|---|---|
| **Comment 8:**
**Line 433-435: I am not sure this analysis fully supports this conclusion: "demonstrating evidence for the pulse-reserve hypothesis and suggesting sub-weekly (rain pulse) rather than seasonal phenological controls on growth (Noy-Meir, 1973)". As the authors have demonstrated, plant growth with longer tp periods are associated with wetter preceding conditions and stronger pulses. This could be in seasons that are already favorable for growth (as the authors state in lines 368-369), suggesting seasonal phenological controls (which may include temperature constraints) are still important. The pulses just result in that extra bit of growth.** | This is an excellent point. As the reviewer points out, we do acknowledge that there is a phenological component to the growth in lines 418-420. Therefore, the statement about phenological controls in original manuscript line 433 is a misstatement of the results. We have slightly revised a similar statement in lines 407-408 and changed line 482-483 to be more consistent with our discussion statements that are similar to the point made by the reviewer. |
| **Comment 9:**
**Given the studies that show inter-annual variability in net CO2 uptake is strongly linked to days with peak gross CO2 uptake (Zscheischler et al., 2016), I am wondering whether increases in leaf growth during the longer tp periods translate to increases in carbon uptake. Perhaps SIF data would be useful in this regard. However, this is probably beyond the scope of this study.** | While evaluating carbon fluxes/photosynthesis is beyond the scope of this study, we have another study in press that we have referenced in lines 414-416 and lines 490-491. The reference is below. In that paper, we show that carbon fluxes at flux towers show similar signatures to VOD responses here, specifically that the greatest and longer duration increases in net ecosystem production occur after larger rainfall events on wetter surfaces.

Feldman, A.F., Chulakadabba, A., Short Gianotti, D.J., Entekhabi, D., 2020a. Landscape-scale plant water content and carbon flux behavior following moisture pulses: from dryland to mesic environments. Water Resour. Res. In Press. |
| **Comment 10:**
**Minor comments**
**Fig. 4C is not referenced in the text.** | Good catch. We have mentioned 4C (now 5C) in the parentheses in addition to 5A and 5D in line 270. We have also denoted 5C in the parentheses in line 426. |
| **Comment 11:**
**Fig. S5: describe sub-figure C in caption.** | Thank you for pointing this out. In Fig. S6 (originally Fig. S5) caption, we have added: "(C) Incorporating random noise into the algorithm appears to increase false detection of non-zero $t_p$ the most. Ultimately, all effects together still result in frequent correct detection of true $t_p$ of zero." |